# Structural Disorder of Graphite and Implications for Graphite Thermometry

Martina Kirilova[1], Virginia Toy[1], Jeremy S. Rooney[2], Carolina Giorgetti[3], Keith C. Gordon[2], Cristiano Collettini[3], Toru Takeshita[4]

[1] Department of Geology, University of Otago, PO Box 56, Dunedin 9054, New Zealand

[2] Department of Chemistry, University of Otago, PO Box 56, Dunedin 9054, New Zealand

[3] Dipartimento di Scienze della Terra, Università degli Studi La Sapienza, Rome, Italy

[4] Faculty of Science, Earth and Planetary Sciences, Hokkaido University, Sapporo, Japan.

*Correspondence to:* Martina Kirilova (martina.a.kirilova@gmail.com)

**Key Points:**

graphite, disorder, thermometry, Raman

**Abstract**

Graphitization, or the progressive maturation of carbonaceous material, is considered an irreversible process. Thus, the degree of graphite crystallinity has been calibrated as an indicator of the peak metamorphic temperatures experienced by the host rocks. However, discrepancies between temperatures indicated by graphite crystallinity versus other thermometers have been documented in deformed rocks. To examine the possibility of mechanical modifications of graphite structure and the potential impacts on graphite 'thermometry' we performed laboratory deformation experiments. We sheared highly crystalline graphite powder at normal stresses of 5 and 25 MPa and aseismic velocities of 1 µm/s, 10 µm/s and 100 µm/s. The degree of graphite crystallinity both in the starting and resulting materials was analyzed by Raman microspectroscopy. Our results demonstrate consistent decrease of graphite crystallinity with increasing shear strain. Microstructural observations show that brittle processes caused the documented structural disorder of graphite. We conclude that the calibrated graphite 'thermometer' is ambiguous in active tectonic settings and we suggest that a calibration that accounts for shear strain is needed.

## 1. Introduction

Organic matter, preserved in sedimentary rocks, can be transformed into crystalline graphite due to structural and compositional changes during diagenesis and metamorphism, a process known as graphitization (Bonijoly et al., 1982; Wopenka and Pasteris, 1993; Beyssac et al., 2002a; Buseck and Beyssac, 2014; etc.). Graphitization is thought to be an irreversible process and graphite is known to remain stable to the highest temperatures of granulite facies and the highest pressures of coesite-eclogite facies (Buseck and Beyssac, 2014). It is generally accepted that the degree of graphite crystallinity, or its structural order, is determined mainly by the maximum temperature conditions experienced by the host rocks, whereas lithostatic pressure and shear strain are considered to have only minor influence on graphitization (Bonijoly et al., 1982; Wopenka and Pasteris, 1993; Bustin et al, 1995). Therefore, graphite crystallinity has been calibrated as an indicator of the peak temperatures reached during progressive metamorphism (Beyssac et al., 2002a; Reitmeijer and McKinnon, 1985). However, in strained rocks discrepancies between temperatures indicated by the crystallinity of graphite vs. other thermometers have been reported (Barzoi, 2015; Nakamura et al., 2015; Kirilova et al., in press). Thus, numerous authors have speculated that tectonic deformation results in graphite structural modifications that challenge the validity of the existing graphite thermometers (Large et al., 1994; Bustin et al, 1995; Crespo et al., 2006; Barzoi, 2015; Nakamura et al., 2015).

Furthermore, graphite occurrence and enrichment have been documented in several fault zones in the world, e. g. the Alpine Fault zone, New Zealand (Kirilova, et al., in press), the Hidaka metamorphic belt, Hokkaido, Japan (Nakamura et al., 2015), the Atotsugawa fault system, Japan (Oohashi, et al., 2012), the Tanakura Tectonic Line, Japan (Oohashi et al., 2011), the Err nappe detachment fault, Switzerland (Manatschal, 1999), and the KTB borehole, Germany (Zulauf et al., 1990). In these intensely deformed rocks its presence is of particular interest because its low friction coefficient of µ ~ 0.1 (Morrow et al., 2000) allows graphite to act as a natural solid lubricant (Savage, 1948). The mechanical behavior of graphite has been broadly investigated in both natural and experimental specimens, where it manifests with the lowest µ among sheet structure minerals (Moore and Lockner, 2004; Oohashi et al., 2011, 2013; Rutter, et al., 2013; Kuo et al., 2014, etc.) confirming it could have a significant impact on fault mechanics. It has been experimentally proven that even a small fraction of graphite can have a disproportionally large effect on frictional strength where graphite is concentrated by smearing into interlinked layers (Rutter, et al., 2013).

However, structural changes in crystalline graphite caused by tectonic deformation have not yet been systematically explored. To examine this aspect and to investigate the potential impacts of structural disordering of graphite on the graphite 'thermometer', we have carried out laboratory deformation experiments on highly crystalline graphite powder.

## 2. Experimental methods

### 2.1 Sample description

As a starting material in the current study we used synthetic (commercially synthesized) graphitic carbon to avoid complexities arising from variable degree of crystallinity in natural carbon materials. Initially, the material was crushed to maximum grain size of 160 µm in a RockLabs Swing (TEMA) mill. The resulting fine graphitic powder was annealed at 700˚C for two hours in a Lindberg Blue M Muffle Furnace to achieve full graphitization, which is known to occur at this temperature in the absence of other variations in physical conditions (Buseck and Beyssac, 2014). This was used as the starting material for the deformation experiments.

### 2.2 Experimental procedure

In total, 10 deformation experiments were performed at room temperature and room humidity in the Brittle Rock deformAtion Versatile Apparatus BRAVA (Collettini et al., 2014), at INGV, Rome. For each experiment two 3-mm thick layers of synthetic graphite gouges were placed in between three grooved forcing blocks in a double-direct shear configuration (e.g. Dieterich, 1972). The two side blocks are held stationary, and the central forcing block is driven downward causing shear to occur within the graphite gouge layers. Normal stress is applied by the horizontal piston in load-feedback control mode and shear displacement accomplished by the vertical piston in displacement-feedback control mode. Forces are measured with stainless steel load cells (± 0.03 kN) and displacements are measured with LVDTs (± 0.1 µm) attached to each piston. Experiments have been conducted at normal stresses of 5 MPa or 25 MPa and aseismic sliding velocities of 1 µm/s, 10 µm/s and 100 µm/s. The experiments were carried out to total displacements of 20 mm. In addition, some experiments were stopped at 5 mm and 10 mm and the specimens were then recovered to reveal graphite structural changes that took place during different amounts of total deformation. The coefficient of friction (µ) was calculated as the ratio of measured shear load to measured normal load ($\mu = \tau / \sigma_n$, where $\tau$ is shear stress and $\sigma_n$ is effective normal stress). The average shear strain within the layer was calculated by dividing shear displacement increments by the measured layer thickness and summing. The displacement values of the vertical and horizontal load points were corrected for the elastic stretch of each load frame, taking into account that the machine stiffness is 1283 kN/mm on the horizontal axis and 928.5 kN/mm on the vertical axis. In addition, we calculated total frictional work for each experiment as a function of shear stress integrated over the total displacement (Beeler, 2007).

### 2.3 Raman microspectroscopy

The degree of graphite crystallinity was measured by an Alpha 300R+ confocal Raman microscope (WITec, Ulm, Germany) with a 532 nm laser (Coherent, Santa Clara, California), located at the Department of Chemistry, University of Otago, New

Zealand. The laser (3.0 mW) was focused on the samples with a 50× Zeiss objective. The scattered light was dispersed with
a 1200 g/mm grating. The combination of the 50× objective and 532 nm laser wavelength produced a laser spot size of
approximately 412 nm in diameter. The integration time of each spectrum was 2 seconds with 50 co-additions (100 seconds
in total). The spectra were calibrated using the Raman band from a silicon wafer prior to each set of measurements.
The collected spectra were pre-processed in GRAMS AI 9.1 (Thermo Fisher Scientific Inc.), where cosmic spikes were
removed and a multi-point linear baseline offset was performed. This was followed by peak fitting three Lorentzian-
Gaussian functions to each spectrum with a linear baseline over 1000 - 1700 $cm^{-1}$. For each spectrum, the area ratio was
calculated ($R2 = A_{D1} / (A_G + A_{D1} + A_{D2})$, where $Ai$ is the area of the $i$th peak, G band is the main high frequency band of
graphite, D1 and D2 bands are defect bands observed in the first order Raman spectrum of graphite) (Wopenka and Pasteris,
1993; Beyssac et al.., 2002a).
**2.4 Scanning electron microscopy**
Microstructural analyses of the graphite gouge recovered from the biaxial apparatus were carried out using a scanning
electron microscope (SEM). Some SEM images were acquired from the shiny surfaces of the graphite layers that had been
parallel to the center and or side forcing blocks (Y-Z sections), with a Zeiss Sigma field emission scanning electron
microscope (VP FEG SEM) at the Otago Centre for Electron Microscopy (OCEM), University of Otago, New Zealand. The
instrument was operated in variable pressure mode (VP) at 15 kV using a working distance (WD) of 7 − 8 mm and a VPSE
(VP-mode secondary electrons) detector. In addition, polished thin sections cut perpendicular to the surface of contact with
the center and side forcing blocks (X-Z sections) were imaged on a JEOL JSM-6510 SEM at the University of Potsdam,
Germany, where high-resolution secondary electron images were collected at 20 kV and a WD of 10 mm.
**2.5 Transmission electron microscopy**
Transmission electron microscopy (TEM) was used for detailed microstructural characterization of the shiny surfaces. High-
resolution TEM images were collected by using a JEM-2010 electron microscope, located at the University of Hokkaido,
Sapporo, Japan. The instrument was operated at 200 kV with LaB6 filament. TEM foils (with size of 12 x 5 µm and
thickness of 1 µm) milled by FIB perpendicular to the shiny surface (X-Z section) were placed on a carbon coated film, and
examined by using dual-axis tilting holder.

## 3. Results

### 3.1 Mechanical data

Our experiments allowed us to investigate graphite mechanical behavior and structural modifications under various sliding velocities, normal stresses and shear strain. These conditions are summarized in Table 1.

### 3.1.1 Friction variations

Over several mm of displacement, the friction coefficient shows a similar evolution trend in all experiments. On a plot of friction coefficient vs. displacement (Fig. 1a), the friction coefficient ($\mu$) delineates a curve characterized by a rapid increase to an initial peak friction coefficient ($\mu_{peak}$), followed by a subsequent exponential decay towards a steady-state friction coefficient ($\mu_{ss}$) over a slip weakening distance. The shapes of the friction-displacement curves vary with the normal stress applied and are steeper for the experiments conducted at 25 MPa than the ones at 5 MPa (Fig. 1a) i.e. the displacement required to achieve steady-state decreases at higher normal stress. In addition, the values of both $\mu_{peak}$ and $\mu_{ss}$ (Fig. 1a; Table 1) are significantly lower in the experiments at 25 MPa ($\mu_{peak}$ = ~0.4; $\mu_{ss}$ = ~ 0.1) than in the experiments at 5 MPa ($\mu_{peak}$ = ~0.5; $\mu_{ss}$ = ~ 0.2) (where $\mu_{ss}$ values were read at the end of each experiment). Plots of $\mu$ at all sliding velocities (Fig. 1a) show subtle variations in $\mu_{peak}$ and $\mu_{ss}$ with change of the applied sliding velocities (Fig. 1a; Table 1).

### 3.1.2 Shear strain variations

Plots of friction coefficient vs. shear strain (Fig. 1b) show significant variations in shear strain attained over equivalent sliding displacements. The estimated shear strain values are a geometric consequence of different thickness changes. Consideration of the shear strain at equivalent sliding velocities but different normal stresses demonstrates that shear strains achieved during the 5 MPa experiments are approximately half of those at 25 MPa (Fig. 1b; Table 1). In addition, the experiments at 25 MPa demonstrate a dramatic increase in shear strain with increasing slip velocity (Fig. 1b; Table 1), whereas at low normal stress we do not observe any systematic variations associated with changes in sliding velocities (Fig. 1b, c and d). Fig. 1c and d show the experiments at low shear strain used to characterize graphite structural changes in the early stages of deformation (Table 1).

### 3.2 Graphite crystallinity

All the experiments resulted in the development of shiny smooth surfaces with gentle slickenlines (macroscopic fine grooves, parallel to the slip direction as defined by Toy et al., in press). Raman spectra obtained on the top of these surfaces, that had accommodated most of the induced deformation, are compared to Raman spectra from the starting material to identify the effects of mechanical deformation on graphite crystallinity.

Raman data from 20 spectra per sample are presented in Supplementary material 1 (S1). Representative spectra for each sample are illustrated in Fig. 2, which shows spectra displaying the least (left column) and the most (right column) disordered graphite within a sample (i.e. lowest and highest R2 values respectively). Spectra that were typical of the average for each sample are also presented (middle column). Experiments 3 and 7 were stopped at only 5 mm displacement and resulted in extremely fragile deformed surfaces, which were unable to be extracted without them breaking into pieces too small to obtain spectra from. Thus, graphite crystallinity was not measured in these experiments.

All the acquired spectra show typical G, D1 and D2 bands, respectively at ~1580 cm$^{-1}$, ~1350 cm$^{-1}$ and ~1620 cm$^{-1}$ (S1). The degree of graphite crystallinity in each sample could thus be calculated by using the area ratio R2 (Fig. 2; S1). Raman spectra collected from the starting material show R2 values ranging from 0 to 0.327 (Fig. 2), corresponding respectively to fully crystalline and highly organized graphite. Spectra acquired from the deformed surfaces show higher R2 values (Fig. 2; S1). The most crystalline graphite with R2=0.330 was collected in Exp. 2 (Fig. 2) while the most disordered graphite with R2=0.661 resulted from Exp. 10 (Fig. 2).

As graphite crystallinity varies within a sample (Fig. 2; S1), we examine average R2 values for each one and compare them with applied normal stress, sliding velocity, shear strain, and total frictional work (Table 2). The starting material has average R2$_{pre-shear\ graphite}$ = 0.173, whereas all deformed samples have higher average R2 values (Table 2). Analyzing the average R2 values for deformed samples reveals that graphite is more disordered in the high normal stress experiments (Table 2) than in the experiments at 5 MPa. Furthermore, in the experiments at 25 MPa the average graphite crystallinity decreases with increasing sliding velocities (Table 2). In contrast, at low normal stress, we do not observe any dependence of the degree of graphite crystallinity on the applied sliding velocities (Table 2). Overall graphite appears as most disordered in the experiments where the highest shear strain was achieved (Table 2). The relationship between average R2 and shear strain is illustrated in Fig. 3a by fitting a power function with a correlation coefficient $R^2$ = 0.95. Fitting a power function to average R2 and total frictional force showed a consistent correlation (Fig. 3b). The experiments 2 and 6 at low normal stress, which were stopped at 10 mm displacement and accommodated the least amount of shear strain, contain the least disordered graphite (Fig. 3; Table 2).

### 3.3 Microstructural characteristics

### 3.3.1 Scanning electron microscopy (SEM)

Similar microstructural features were observed in all the deformed samples. SEM images obtained from the sample deformed during experiment 8 are presented to demonstrate our observations (Fig. 4).

These high-resolution images in Y-Z sections reveal that the shiny surfaces are decorated by closely spaced (from < 5 to 10 micrometers) slickenlines (Fig. 4a), on top of a smooth continuous layer. In places, the continuity of this layer is interrupted by fine (~1 to 2 micrometers in width) fractures (Fig. 4a), with random orientation compared to the slip direction. Occasionally, the deformed surface appears as completely disrupted, and is decorated with smaller graphite grains from 10 to 50 micrometers in size, oriented nearly parallel to the shear direction (Fig. 4b). In X-Z sections this highly deformed surface is observed as a thin slip-localized zone, composed of well-compacted layer of aligned graphite grains (Fig. 4c). This localized shear surface is underlain by a zone of randomly oriented, inequigranular, irregular graphite grains (Fig. 4d). In places, most of the graphite grains are aligned with their basal (001) planes parallel to the slip direction, and form compacted layers, defining a weakly-developed fabric (Fig. 4e). There has been some dilation along these cleavage planes, and the spaces thus created are filled with smaller graphite grains with their (001) planes sub-perpendicular to the shear direction (Fig. 4e). Locally, intensely fractured grains are also observed (Fig. 4f).

**3.3.2 Transmission electron microscopy (TEM)**

TEM was used to examine the microstructure of the material that makes up the shiny surfaces (Fig. 4c). TEM analyses were performed on foils cut perpendicular to this surface. Fig. 5 shows characteristic TEM images obtained from the sample recovered from experiment 8.

Graphite grains in this well-compacted layer have basal planes predominantly aligned with the shear plane, as were observed in SEM images. However, adjacent grains show slightly different orientations (Fig. 5a). In addition, kink folded graphite grains are observed in multiple locations in the foils (Fig. 5b, c), which yields a 'wavy layering' at a small angle to the shear direction (Fig. 5b). In isolated areas, there are also some smaller grain fragments with random orientation (Fig. 5d).

**4. Discussion**

**4.1 Mechanical behavior**

Graphite in our experiments shows mechanical behavior consistent with other mechanical studies of pure graphite gouges. Our results display low $\mu_{ss}$ values (from ~0.1 to ~0.2; Table 1) as did the low-pressure deformation experiments of carbonaceous material performed by Morrow et al. (2000), Moore and Lockner (2004), Oohashi et al. (2011, 2013), Kuo et al. (2014), and Rutter et al. (2013). The low frictional strength of graphite is well known and has been attributed to its sheet structure composed of covalently bonded carbon atoms held together only by van der Waals forces. These weak interlayer bonds along (001) planes are easily broken during shear (Moore and Lockner, 2004; Rutter, et al., 2013). Initial $\mu_{peak}$ followed by strain weakening during deformation experiments of graphite gouges has been previously explained with the

work involved in rotating the grains with their (001) planes sub-parallel to the shear surfaces, which puts them in the optimal
position for shearing along the weak interlayer bonds (Morrow et al., 2000; Moore and Lockner, 2004; Rutter, et al., 2013).
Controversially, Oohashi et al. (2011) reported an absence of $\mu_{peak}$ in pure graphite gouges sheared at $\leq 2$ MPa with sliding
velocities of 1.3 m/s. Instead shearing started and continued at a similar $\mu$ throughout their experiments. We hypothesize that
higher velocities result in more efficient reorientation of graphite grains, and therefore, $\mu_{peak}$ is not present in experiments
carried out at seismic rates. We also acknowledge that the imposed velocities in the experiments by Oohashi et al. (2011)
were substantially different to ours, and shearing at those seismic rates may cause frictional heating. Therefore, graphite
frictional strength in their experiments may be related to thermally-activated weakening mechanisms (Di Toro et al., 2011)
that are only significant at these high velocities.
We also observed shear strain variations in the various samples (Fig. 1b, c and d) that are systematically related to the
conditions of the experiments. The calculated shear strain (or the ratio of shear displacement to measured layer thickness) is
directly dependent on the applied normal stress, and shear strains are significantly higher in the experiments performed at 25
MPa than the ones at 5 MPa due to better compaction and thinning of the sheared graphite gouges. Furthermore, sliding
velocities also play a role in the accommodated total shear strain, and shear strain increases with increase in the applied
sliding velocities but only in the high normal stress experiments (Fig. 1b). As we previously suggested, higher velocities may
result in more efficient reorganization of graphite grains, and thus further progressive thinning of the graphite gouges
occurred. However, we cannot explain the absence of similar trend at the 5 MPa experiments by our results. There are too
few of these relationships to fully characterize the effect of sliding velocity on shear strain accumulation in graphite gouges,
and more mechanical data of this sort need to be collected in future.
**4.2 Structural disorder of graphite**
Our experimental study clearly demonstrates transformation of fully/highly crystalline graphite (with R2 ratios ranging from
0 to 0.327; Fig. 2; S1) into comparatively poorly organized graphitic carbon (with R2 ratios up to 0.661; Fig. 2; S1), which
indicates significant graphite disorder with increasing strain and total frictional work at the tested aseismic sliding velocities
(Fig. 3). The estimated bulk shear strains (Table 1) are likely to be significantly lower than the shear strains accommodated
within the thin shear surfaces. However, we expect the strain variations within these surfaces to be directly related to the
measured bulk shear strains. Nevertheless, we refer to the above relationship as a rough approximation. We also
acknowledge that the slickenlined surfaces that were produced experimentally contain some graphite that yield spectra
comparable to those acquired from the starting material i.e. there is highly crystalline graphite that appears as unaffected by
the deformation. However, at least some of these spectra are derived from undeformed graphite powder that underlies the
shear surfaces and could not be entirely removed during sample preparation due to the fragile nature of the samples. It is also
possible that some non-deformed graphite powder was accidently measured through the fractures that are cross-cutting the

accumulated shear surfaces (Fig. 4a). But even if some graphite did not undergo mechanical modification during the experiments, the results overall validate that structural disorder of graphite can result from shear deformation subsequent to the graphitization process.

We evaluate the documented disorder of the crystal structure of graphite by analyzing variations in R2 ratios, which depend on the increase of defect bands (D1 and D2) in the Raman spectrum of graphite. This relationship is a well-known crystallinity index of graphite that shows the degree of maturity of the carbonaceous material (Wopenka and Pasteris, 1993; Beyssac et al., 2002a; etc.). Alternatively, it may reflect increase in the grain boundary density (Tunistra and Koening, 1970; Pimenta et al., 2007). However, we aimed to avoid grain boundaries during spectra acquisition, which was possible due to the laser spot size of 412 nm, which is much smaller than the graphite grains in our samples (>10 microns, Fig. 4b). We acknowledge that some of the spectra may have been accidentally obtained in close proximity to grain boundaries, however occasional measurements of this sort are unlikely to affect the average R2 per sample. Thus, we attribute the detected increase in D bands in our experimental data to disorder of the internal structure of graphite rather than grain size reduction.

Our findings contradict the paradigm that the degree of graphite crystallinity is determined by an irreversible maturation of carbonaceous material (Bonijoly et al., 1982; Wopenka and Pasteris, 1993; Beyssac et al., 2002a; Buseck and Beyssac, 2014). Therefore, graphite should not be considered as a stable mineral (Buseck and Beyssac, 2014), especially in active tectonic settings, where mechanical motions, such as fault creep, may cause disordering of the structure of carbonaceous material that formed during typical graphitization processes. Similar assumptions have been made on graphite in intensely deformed cataclasites (comprising crushed mylonitic chips floating in a fine-grained matrix) that is significantly disordered in comparison with graphite in the spatially associated mylonitic rocks (Kirilova et al, in press; Nakamura et al., 2015).

We have experimentally proven that shear strain can not only affect the final structural order of graphite but also manifests as a controlling parameter in the transformation process (Fig. 3a; Table 2). Previous authors have suspected that shear strain may play an important role for graphite modifications, and evidence for this has been found in graphite crystallinity variations in natural samples from active fault zones (Kirilova et al, in press; Nakamura et al., 2015), and strained rocks in metamorphic terrains (Barzoi, 2015; Large et al., 1994). Thus, we conclude that the previously proposed model of progressive graphitization due to increase of temperature (Bonijoly et al., 1982) does not completely reflect the graphite formation mechanisms.

Furthermore, graphite can form or be transported at various depths by tectonic processes, and therefore, it can be exposed to different lithostatic pressures, and hence different normal stresses. We demonstrated that during shearing higher normal stress results in an increase of shear strain (Fig. 1b), and thus causes a higher degree of graphite disorder (Fig. 3a; Table 2). This outlines the significant effect of lithostatic pressure on graphite crystallinity that has been undervalued until now (Bonijoly et al., 1982; Wopenka and Pasteris, 1993; Bustin et al, 1995; Beyssac et al. 2002b). Previous experimental studies

have identified initiation and enhancement of graphitization under pressure (i. e. increase in graphite crystallinity) but only at nanometer scale (Bonijoly et al., 1982; Beyssac et al., 2003). Nevertheless, we speculate pressure should be also considered as a factor that can determine the degree of graphite crystallinity during both graphitization and graphite structural modifications.

We have investigated the effects of shear strain and pressure on graphite crystallinity during shear deformation with aseismic velocities, using a starting material with uniform properties (i.e. highly crystalline graphite powder). In contrast, Kuo et al. (2014) and Oohashi et al. (2011) simulated fault motions in synthetic and natural carbonaceous material with variable degree of maturity at the start of the experiments (ranging from amorphous carbonaceous material to crystalline graphite). Both studies reported graphitization of carbonaceous material due to localized frictional heating rather than structural disordering. These experiments reveal the impact of seismic velocities on graphite structural order and the fact their findings differ so markedly from ours highlights the complexity of graphite transformations in fault zones.

Our microstructural observations provide some indications of the deformation processes that affected graphite structural order. The shiny slickenlined surfaces are composed of very fine-grained material visible as slip-localized zone on SEM images (Fig. 4d). Nanoscale observations reveal graphite grains within it occasionally form stacked kink-band structures, (Fig. 5b, c). This zone, which we assume accommodated most of the induced deformation, is underlined by a less deformed zone composed of larger graphite grains in a finer matrix that in places has developed as an anastomosing fabric, typical of creeping gouges (Fig. 4d). In rare places at SEM scale brittely fractured grains also occur (Fig. 4f and 5d). The interpreted structures suggest that brittle processes operated during shearing, and we conclude that these processes resulted in the structural disorder of graphite, manifested as changes in the Raman spectra. This interpretation is in agreement with the conditions of our experiments (i.e. shearing with aseismic velocities took place at room temperature conditions), that typically would not induce temperatures high enough for crystal plastic processes. Furthermore, the microstructures and the inferred processes are exactly the same as those observed by Nakamura et al. (2015) in the Hidaka metamorphic belt, Japan.

However, crustal fault zones do not only accommodate brittle deformation. At higher temperatures and confining pressures, localised shearing can operate by plastic mechanisms (White et al., 1980). We hypothesize that graphite crystallinity could also be influenced by plastic deformation, as was also suggested in previous studies by Large et al. (1994), Bustin et al. (1995), Barzoi et al. (2015). Investigating this hypothesis and identifying the exact effects of strain on graphite crystallinity during ductile deformation remain goals for future research.

**4.3 Implications for graphite thermometry**

The crystallographic structure of graphite measured by Raman spectroscopy has been applied as a thermometer that relies on progressive maturation of originally-organic carbonaceous material during diagenesis and metamorphism. Previous studies

have focused on calibrating this thermometer. The current best calibration is described by the following equation T (ºC) = - 445 * R2 + 641 ± 50 (Beyssac et al. 2002) by inferring a linear correlation between R2 ratio and peak metamorphic temperatures. However, this thermometer disregards the effects of mechanical modifications of the graphite structure, which this study has identified as having a substantial influence on graphite crystallinity in deformed rocks at sub-seismic velocities.

Our experiments demonstrate a shear strain-dependent increase of the R2 ratio of initially highly crystalline graphite powder due to brittle deformation (Fig. 3a; Table 2). In natural analogues, the pre-shear graphite would yield temperatures up to 641 ± 50 ºC (S1), which is the upper limit of the calibrated thermometer (Beyssac et al. 2002). Whereas, the sheared samples would indicate peak metamorphic temperatures as low as 347 ± 50 ºC (estimated from the most strained samples; S1). Thus, we experimentally prove that in active tectonic settings graphite thermometers may underestimate the peak metamorphic temperatures by < 300 ºC. In cataclasites from the Alpine Fault zone, New Zealand (Kirilova et al., in press) and fault zones of the Hidaka metamorphic belt, Japan (Nakamura, et al., 2015), the graphite thermometer yields temperature discrepancies of more than 100 ºC compared to temperature estimates derived both from the surrounding high-grade amphibolite facies mylonites and the lower grade equilibrium cataclastic phases (marked by chlorite alteration). Barzoi (2015) also described differences of ~ 150 ºC in graphite temperatures between strained and less strained low grade metamorphic rocks from Parang Mountains, South Carpathians.

We conclude that shear strain calibration of the current graphite thermometer is needed, and we propose an appropriate adjustment based on our dataset. Fig. 3a illustrates good correlation between the average R2 and the bulk shear strain measured within a sample, which can be described by the following equation (1):

$F(x) = 0.14017 * x^{0.30713} + 0.15629$ with a correlation coefficient $R^2 = 0.95$         (1)

where x = bulk shear strain.

 However, a calibration of the existing graphite thermometer could be still insufficient to permit reliable temperature estimates in active tectonic settings because both aseismic and seismic sliding velocities are likely to be encountered in fault zones, resulting in structural disorder of graphite or graphitization (Oohashi et al., 2013) respectively. Furthermore, it can be challenging to estimate shear strain in natural samples, so a strain-calibrated graphite thermometer may be impossible to use in deformed rocks.

**5. Conclusions**

We have experimentally demonstrated that graphite crystallinity can be reduced by deformation by performing shear deformation experiments at aseismic sliding velocities insufficient to generate appreciable frictional heat on graphite gouges

composed of powdered highly-organized graphite. Our results clearly demonstrate significant decrease in graphite structural order, which is a function of the total shear strain attained during the various experiments. Microstructural data presented here reveal that this is a result of brittle processes. We also observed a trend of increasing shear strain within a sample with increase in the applied normal stresses and sliding velocities. This reveals the complexity of graphite structural modifications and highlights the significance of the various parameters that can affect the graphitization process. Our findings compromise the validity of the calibrated graphite thermometer by showing they may underestimate the peak metamorphic temperatures in active tectonic settings. We further suggest a simple shear strain calibration of this thermometer.

## Acknowledgments

The research was funded by the Department of Geology, University of Otago, New Zealand, and Rutherford Discovery Fellowship RDF-UOO0612 awarded to Virginia Toy. We also acknowledge the 'Tectonics and Structure of Zealandia' subcontract to the University of Otago by GNS Science (under contract C05X1702 to the New Zealand Ministry of Business, Innovation and Employment). We thank our colleagues Gemma Kerr and Brent Pooley for assistance in sample preparation, and Hamish Bowman for helping with data visualization. We also wish to express our gratitude to Laura Halliday for generously offering to perform grain size analysis on our samples at the Department of Geography, University of Otago, New Zealand. And last but not least, we thank Marco Scuderi for valuable discussions and assistance throughout the experimental procedures.

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

**Table 1.** Summary of the conditions at which experiments were carried out and results.
**Table 2.** Summary of the relationship between shear strain and average R2 within a sample. The conditions of each
experiment are also given as follows: applied normal stress in MPa, sliding velocities in µm/s and sliding displacement in
mm.
**Figure 1**. Plots of mechanical data (a) friction coefficient, µ vs. displacement (b), (c), (d) friction coefficient, µ vs. shear
strain.
**Figure 2.** Representative Raman spectra illustrating: (i) the most crystalline graphite (left column) within a sample; (ii)
graphite with average crystallinity per sample (middle column); and (iii) the most disordered graphite (right column)
encountered in each sample. The R2 ratio for each spectrum is also noted in italic font.
**Figure 3.** Plot of the average R2 ratio vs shear strain accumulated during each experiment.
**Figure 4.** SEM images, obtained from the deformed graphite gouge during experiment 8 (normal stress at 25 MPa with 1
µm/s sliding velocity), show: (a) Slickenlines ornamenting the shear surface; (b), (c) A well-compacted layer of aligned
graphite grains, which make up the shear surface. Bright patches due to a differential charging effect; (d) A less deformed
zone with typical cataclastic fabric, underlying the shear surface; (e) Dilated cleavage planes in large graphite grains filled
with smaller platy graphite grains oriented sub-perpendicular to the shear direction; (f) Fractured graphite grains.
**Supplementary material 1 (S1).** Raman data from 20 spectra per sample together with calculated R2 ratio and average R2
value for each sample. The last column represents temperature estimated by the current best calibration of a Raman-based
thermometer: T ($^\circ$C) = - 445 * R2 + 641 $\pm$ 50.

| Experiment number | Normal stress (MPa) | Sliding velocity (µm/s) | Displacement (mm) | Peak friction coefficient ($\mu_{peak}$) | Steady state friction coefficient ($\mu_{ss}$) | Shear strain maximum |
|---|---|---|---|---|---|---|
| 1 | 5 | 1 | 20 | 0.53 | *0.22* | 17.70 |
| 2 | 5 | 1 | 10 | 0.53 | 0.22 | 8.17 |
| 3 | 5 | 1 | 5 | 0.52 | *not reached* | 4.23 |
| 4 | 5 | 10 | 20 | 0.53 | 0.24 | 20.45 |
| 5 | 5 | 100 | 20 | 0.57 | 0.22 | 16.89 |
| 6 | 5 | 100 | 10 | 0.55 | 0.22 | 9.80 |
| 7 | 5 | 100 | 5 | 0.57 | *not reached* | 3.87 |
| 8 | 25 | 1 | 20 | 0.43 | 0.17 | 21.45 |
| 9 | 25 | 10 | 20 | 0.43 | 0.17 | 31.86 |
| 10 | 25 | 100 | 20 | 0.41 | 0.14 | 46.77 |

**Table 1.** Summary of the conditions at which experiments were carried out and results.

| Sample | Experimental conditions | Shear strain | Average R2 (error estimate ± 0.05) | Total frictional work |
|---|---|---|---|---|
| **Pre-shear graphite** | N/A | N/A | 0.173 | |
| **Exp. 2** | 5 MPa, 1 μm/s, 10 mm | 8.17 | 0.438 | 38.8689 |
| **Exp. 6** | 5 MPa, 100 μm/s, 10 mm | 9.80 | 0.430 | 46.6369 |
| **Exp. 5** | 5 MPa, 100 μm/s, 20 mm | 16.89 | 0.454 | 157.9314 |
| **Exp. 1** | 5 MPa, 1μm/s, 20 mm | 17.70 | 0.506 | 165.4748 |
| **Exp. 4** | 5 MPa, 10 μm/s, 20 mm | 20.45 | 0.517 | 180.8346 |
| **Exp. 8** | 25 MPa, 1 μm/s, 20 mm | 21.45 | 0.520 | 192.9007 |
| **Exp. 9** | 25 MPa, 10 μm/s, 20 mm | 31.86 | 0.580 | 283.7721 |
| **Exp. 10** | 25 MPa, 100 μm/s, 20 mm | 46.77 | 0.604 | 424.0356 |


**Table 2.** Summary of the relationship between shear strain, average R2, and total frictional work within a sample. The conditions of each experiment are also given as follows: applied normal stress in MPa, sliding velocities in µm/s and sliding displacement in mm.

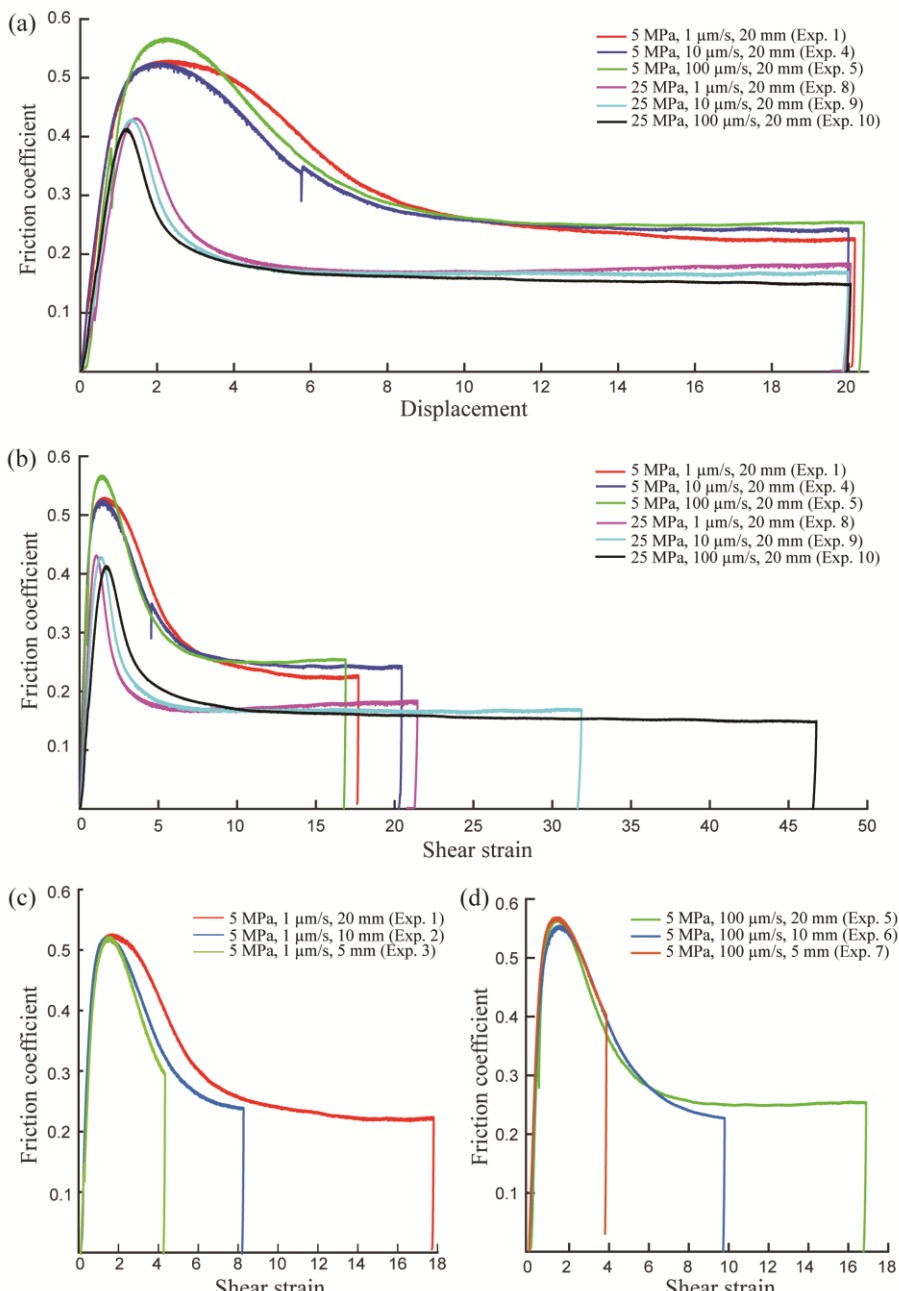

**Figure 1**. Plots of mechanical data (a) friction coefficient, μ vs. displacement (b), (c), (d) friction coefficient, μ vs. shear strain.

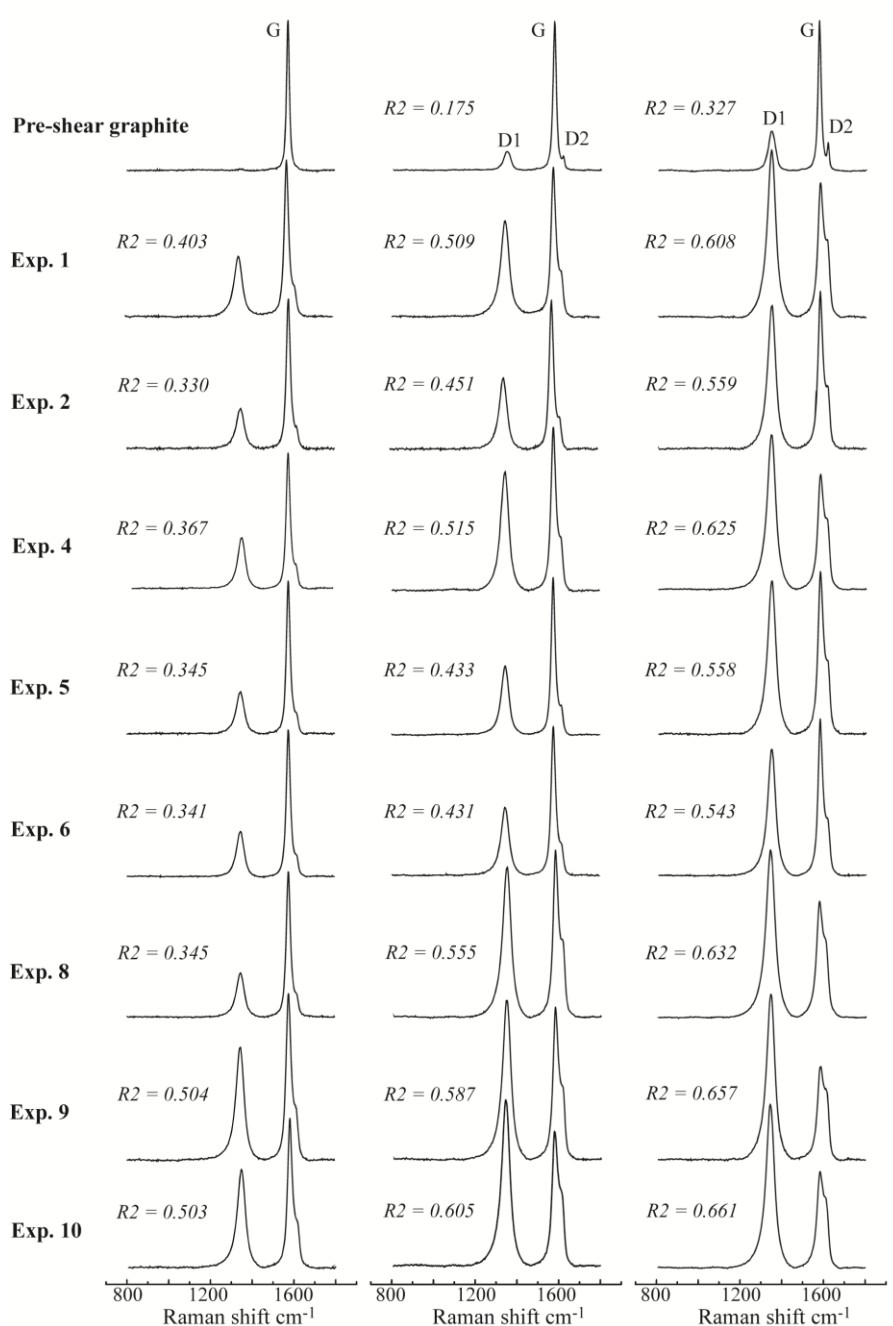

429

**Figure 2.** Representative Raman spectra illustrating: (i) the most crystalline graphite (left column) within a sample; (ii) graphite with average crystallinity per sample (middle column); and (iii) the most disordered graphite (right column) encountered in each sample. The R2 ratio (with an error estimate of 0.05) for each spectrum is also noted in italic font.

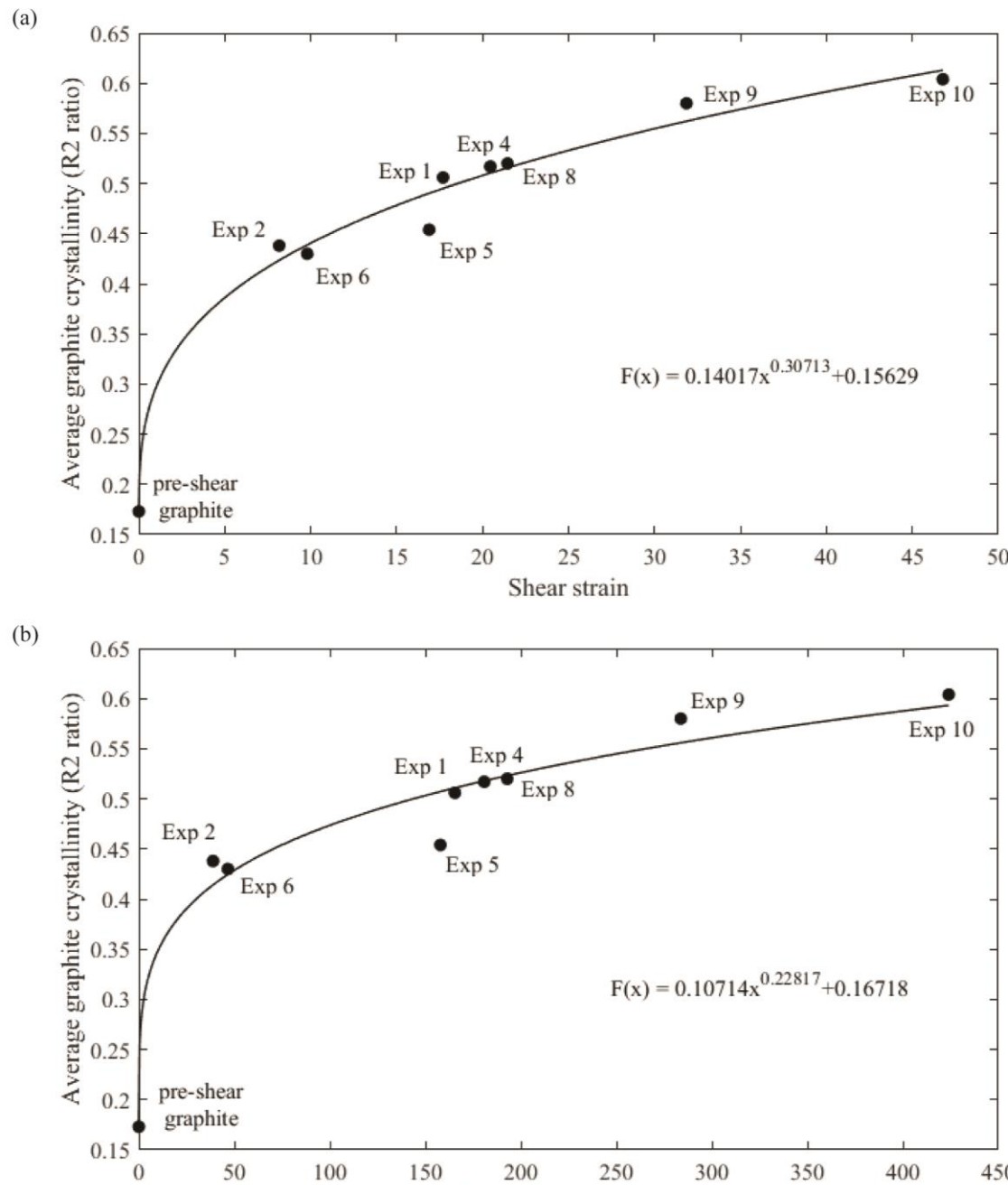

**Figure 3. (a)** Plot of the average R2 ratio vs shear strain accumulated during each experiment. **(b)** Plot of the average R2 ratio vs total frictional work during each experiment.

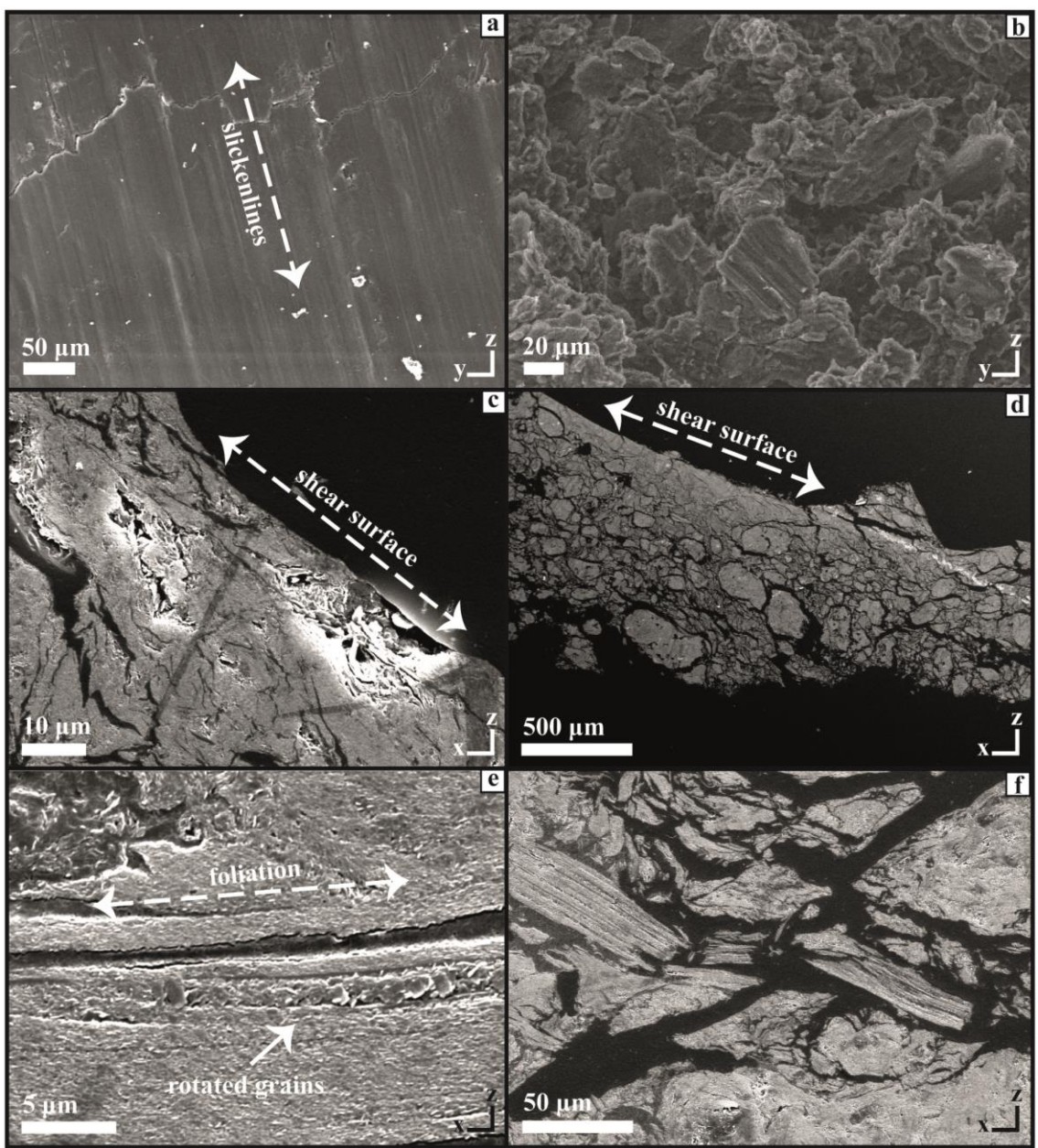

437

**Figure 4.** SEM images, obtained from the deformed graphite gouge during experiment 8 (normal stress at 25 MPa with 1 µm/s sliding velocity), show: (a) Slickenlines ornamenting the shear surface; (b), (c) A well-compacted layer of aligned graphite grains, which make up the shear surface. Bright patches due to a differential charging effect; (d) A less deformed zone with typical cataclastic fabric, underlying the shear surface; (e) Dilated cleavage planes in large graphite grains filled with smaller platy graphite grains oriented sub-perpendicular to the shear direction; (f) Fractured graphite grains.

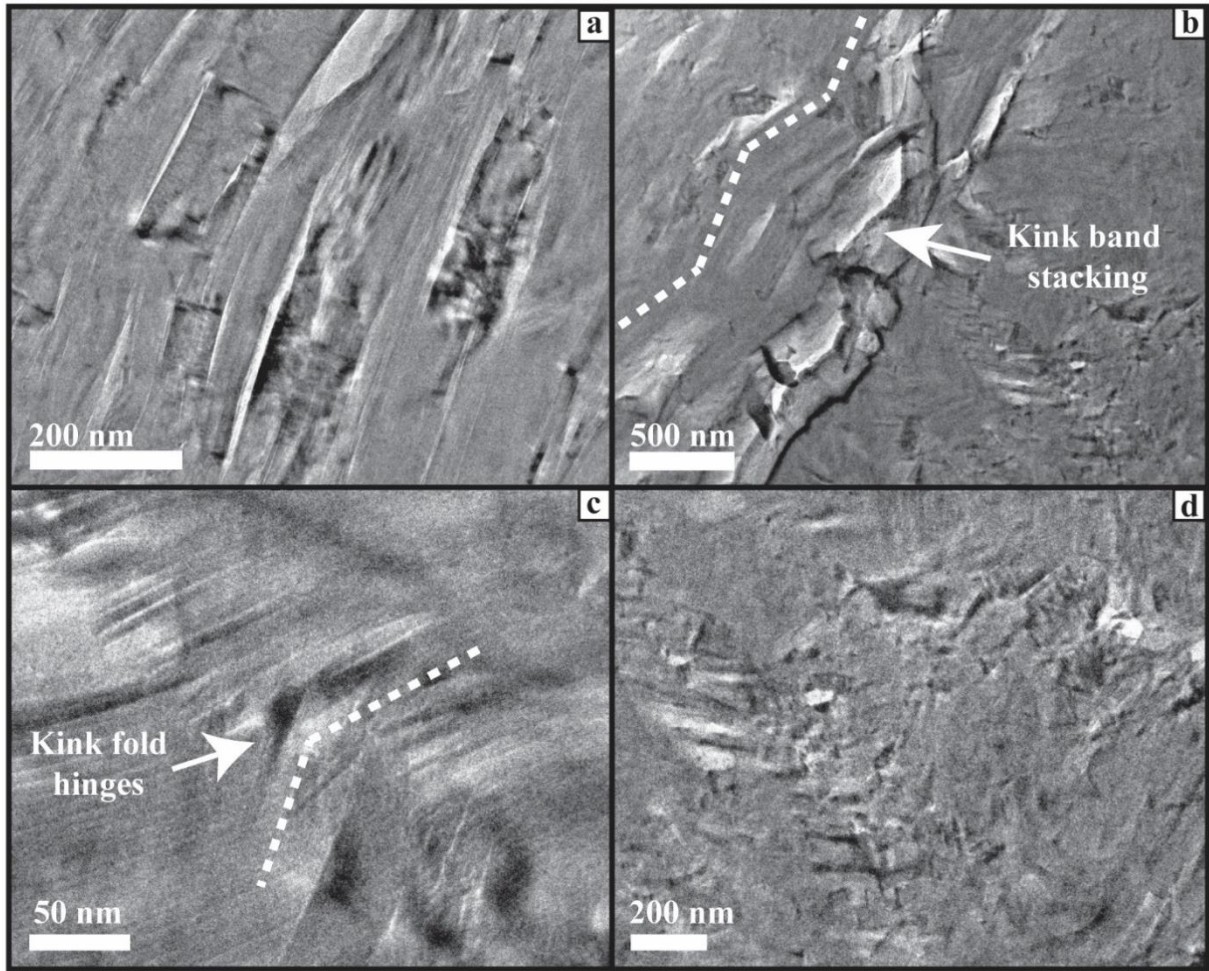

**Figure 5** TEM images showing microstructural characteristics of the slip-localized shear surface: (a) aligned grains showing slightly different orientation; (b) kink band stacking; (c) dilated kink fold hinges; (d) fragmented grains.