# Peer review of "Structural Disorder of Graphite and Implications for Graphite"

_Solid Earth, 2017_

## Referee Comment (RC1) · Dr. Kilian (Referee) · 24 Jul 2017

In the manuscript "Structural Disorder of Graphite and Implications for Graphite Thermometry" s2-2017-74, Kirilova et al. report the results of Raman spectroscopy on an experimentally sheared, synthetic graphite gouge. Raman spectroscopy on carbonaceous material (RSCM) has become a frequently used geothermometer assuming that the crystallinity of graphite represents peak metamorphic conditions. In contrast to geothermometers based on mineral compositions where a retrograde overprint can often be easily recognized, preservation of peak metamorphic conditions under deformation might not be easily recognised in graphite where no compositon changes are to be expected. Accordingly, this contribution is of special interest to researchers, either using RSCM or having to evaluate RSCM derived data. Testing the RSCM thermome-

ter against experimentally deformed graphite gouge is novel and the results of the presented study are very interesting and strongly suggest that the applicability RSCM thermometry to deformed rocks - and accordingly to almost all metamorphic rocks - needs a careful evaluation. This ms highlights that not all parameters to unequivocally interpret RSCM are sufficiently understood. The authors of the ms suggest a correction of the RSCM thermometer based on shear strain, related to their observations in the synthetic graphite gouge.

The manuscript is concise, easy to read and provides a valuable insight into a neglected problem of RSCM. However, there are points that are either not entirely consistent, that need further clarification or corrections, especially in the obtained relation to sample strain and the determination of crystallinity and those should be addressed (listed below). Overall I recommend that the manuscript is suitable for publication after moderate revisions.

Following comments should be addressed.

1) Relation to strain/stress a) A "shear strain" is calculated by summing up the ratios of displacement increments and thickness. As it appear the samples are thinning with increasing displacement, this "shear strain" is neglecting the thinning component, the derived "shear strain" can not be used to calculate a strain ellipse (no functional relation) and overestimates strain in the sample. A more correct procedure would be one where progressive simple and pure shear are treated to occur concurrently, and reporting a unique measure of strain. b) The authors report (and obtain their measurements) from the shiny surface where they assume along which most of the displacement is realised. Assuming that this surface is actually a thin layer, the strain within that layer must always be larger than the strain derived from the entire sample. Hence one might speculate that a functional relation between the determined "shear strain" and R2 is at best a rough approximation. In case, as indicated by some of the comments on the microstructure, the compaction (sample thinning?) is localised as well and not homogeneous, a bulk sample, "strain estimate" is even more likely to be unrelated to the

state of deformation in the analysed layer. c) The authors use a mixture of surface related measures (friction coefficients, normal stress, slip rate) and volume measures (strain) which might be confusing in places, e.g. it might not be directly evident that the experiments with the 25 MPa normal stress should actually be stronger that the 5 MPa experiments.

2) R2 correlation with shear strain a) Following the number in formula 1and as shown in Figure 3., shear shear is evaluated as a function of R2, the inverse would be logical and a fit is numerically not equivalent. Additionally, formula 1 is wrong. b) What is the physical basis that R2 and strain should have a power law relationship?

3) Origin of D1,2 bands Obtaining Raman spectra at surface ledges on (001) or generally grain boundary regions, the appearance of the D bands has been observed (e.g.Tuinstra & Koenig, 1970; Pimenta et al. 2007). For small grain sizes, the ratio of G and D bands is actually used to establish a grain size determination. The authors mention that the increasing area of D1,D2 peaks is related to a decreasing crystallinity of graphite, is it possible that the crystallinity does not change but rather with a smaller grain size more grain boundary area with a disturbed lattice is measured? The authors estimate the minimum resolution of their optical system at 0.4 $\mu$m, however, from the text it becomes not clear whether this relates to the analysed point and/or if the analysed point could actually be identified and if so, whether measurements with a large R2 come from areas with a smaller grain size? If crystallinity is defined in relation to intragranular defect density/lattice perfection, it should be verified that only intragranular measurements are evaluated. If the grainsize is so small that most likely grain aggregates are measured, maybe the grain size effect could be corrected. The authors mention that they actually examine the structural disordering of graphite, so depending how this is defined, it needs to be considered separately to grain boundary effects.

4) Applicability of a strain corrected RSCM thermometer a) in real rocks, it might be difficult to estimate strain and it might not be clear in which way deformation partitions between graphite and other minerals and so I'd encourage the authors to share

some thoughts on how a strain correction should be applicable in a real-rock situation.ĂăÂăÂă b) A correction for the thermometer might depend on the relation of temperature and strain: e.g. during exhumation of a rock deformation takes place at increasingly lower temperature. Using a simple "strain" correction, would imply that the deformation temperatures during exhumation would not have to be considered. Given that most minerals show different deformation mechanisms at different temperatures, it might be reasonable to assume that this is the case as well for graphite. So strictly speaking, a correction should only consider lattice defects introduced by the identical process which is occurring in the calibration experiments. I'd encourage the authors to comment on this complication of such an effort. c) Using any thermometer to determine peak temperatures, often the measurements yielding the highest results are considered as representing peak conditions to overcome the problem of a partial lower temperature overprint. Given that some measurements in deformed samples still yield a low R2, it would be helpful to see where those measurements are actually determined. Are those from within grains while those with large D and D' bands contain areas with a high grain boundary density?

The manuscript and the reader would benefit from some definitions: crystallinity (vs structural ordering for example), interpretation of the peaks/bands in the Raman spectrum, (G being sp2 activation, D1 most likely to intraplane defects and D2 to out-of plane defects, sp3 defects ...) should be introduced and defined. Why are sliding velocity, slip rate, shear velocity are all used synonymously? The figures regarding the microstructural description might benefit from a better resolution since some feature referred to in the text cannot be seen in the figures (comments below).

Minor comments on the manuscript (l=line): l 41:friction coefficient l 52: no "or" l 54: 160 $\mu$m are maximum grain size? l 54: annealed instead of "cooked"? l 85: Please introduce G,D1,D2 l 102: coefficient (please also correct that in the figures if you prefer to stay with friction coefficients instead of shear stress) l 103: coefficient l107: Plots of $\mu$ at all slip rates ..: I do not see the gradual decrease of $\mu$peak (numbers, if I'm not

mistaken are 0.43, 0.43, 0.41, so I wouldn't call this a gradual decrease) l 108 slip rates ...shear velocity: is there any difference? Elsewhere like in the tables there is slip rate, sliding velocity, why are they all used synonymously, please settle for one term. l 110: Where are values of $\mu$ss are read off? l 135: retained instead of produced l 137: varies l 140: high-pressure experiments: should be high normal stress experiments l 152: fractures with random orientation compared to the slip direction: I can't see those l 155: ...well-compacted layer Fig 4c: hard to see in the Figure what is described in the text l 156: "randomly oriented ... Fig 4d": Is that actually confirmed or just based on visual impression l 158: "weak fabric development (Fig. 4e)" I can't see this in the figure l 159: "filled with smaller graphite grains...Fig 4f": also here, it is hard to see, I hope a better resolution of the image and some arrows may help l 176: "more efficient reorientation" What should that be, rotation per time, alignment after strain...? l 178: This clear trend is not so clear to me. l 180: "partial frictional heating": What should that be? l 183 ff: The relation of compaction (volume change), thinning and the apparent dependency of "shear strain" on normal stress and displacement rate should be reconsidered, given that no suitable measure for strain is used l 188: replace high-pressure with high normal stress l 207: "stable mineral" from the thermodynamic point of view it is the stable mineral, so please explain your definition of "stable". l 291-223: There is a mixture of lithostatic pressure and normal stress: Any effect that is observed at higher normal stress does not mean there is a "significant effect of lithostatic pressure". Normal stress is not equivalent to confining/lithostatic pressure. l 234: "... microstructural observations provide some indications of the deformation processes..." These observations could be enhanced! l 239: "plastic mechanisms" Such as? l 240: "plastic deformation" What should that be in contrast to the presented experiments where a gouge seems to flow by a rate independent mechanism? l 244: "crystallographic structure ..." This seems to be a speculation neglecting a grain size effect. l 262ff: Any fit should consider the measure of strain as the independent variable, not as shown and calculated as the dependent one. Independent from that, (1) is not correct. l 273: "We observe a trend..." So it appear there is an increasing thinning which however needs not to translate into

an increased shear strain. l350: "approximate average crystallinity": What should that be?

Figures and Tables:

Table 1 and table 2 are mostly redundant and differ just by 1 column. Figure 1 It would be nice to use identical colors in (c) and (d) representing the displacement. Y-axis should read friction coefficient. Figure 4 could strongly benefit from some images that more clearly show what is described in the text.

Supplementary material: If "int" stands for intensity and "pos" for wavenumber, it seems like the column headers are misplaced.

---

## Referee Comment (RC2) · Dr. Kilian (Referee) · 13 Sep 2017

Reviewer comment:

1) Relation to strain/stress a) A "shear strain" is calculated by summing up the ratios of displacement increments and thickness. As it appear the samples are thinning with increasing displacement, this "shear strain" is neglecting the thinning component, the derived "shear strain" cannot be used to calculate a strain ellipse (no functional relation) and overestimates strain in the sample. A more correct procedure would be one where progressive simple and pure shear are treated to occur concurrently, and reporting a unique measure of strain.

Response:

By "measured layer thickness" we mean the layer thickness measured at the same record number at which the shear displacement increments were measured, i.e. it is the instantaneous layer thickness. Consequently, we take into account the sample thinning with increasing displacement. Perhaps we need to better explain this in the manuscript.

However, we recognize that you have recently recalculated strains in  general shear experiments that take the layer thinning into account (Kilian, R., & Heilbronner, R. 2017, April. Texture transition in experimentally deformed quartzite. In EGU General Assembly Conference Abstracts, Vol. 19, p. 6966). We hope you can advise us in more detail how we could most appropriately make use of such calculations in our current study.

Reply to Response:

Summing up individual steps (delta displacement/current thickness) does not produce a shear strain which accounts for the deformation imposed on the sample. This measure could be termed "apparent shear strain" but it overestimates the true (shear) strain. This apparent shear strain cannot be used to derive a strain ellipse (and accordingly strain related measures i.e. LINFS, ISA, ... any measure of strain). Any correlations of e.g. microstructural features or any other supposedly strain dependent material property should greatly benefit from a correct measure of strain. For a simple procedure to derive bulk strain under the assumption of simultaneous thinning and shearing see e.g. Fossen&Tikoff (1993).

Reviewer comment:

b) The authors report (and obtain their measurements) from the shiny surface where they assume along which most of the displacement is realised. Assuming that this surface is actually a thin layer, the strain within that layer must always be larger than the strain derived from the entire sample. Hence one might speculate that a functional relation between the determined "shear strain" and R2 is at best a rough approximation. In case, as indicated by some of the comments on the microstructure, the compaction (sample thinning?) is localised as well and not homogeneous, a bulk sample, "strain estimate" is even more likely to be unrelated to the state of deformation in the analysed layer.

Response:

We agree with your statement that the measured bulk shear strain is most probably significantly lower than the shear strain accumulated only within the thin shear surfaces. However, we expect the shear strain variations within these surfaces to be linearly correlated with the measured bulk shear strain within a sample. Thus, we believe that a correlation between shear strain and R2 is in fact possible. Nevertheless, we will relate to this relationship as a 'rough approximation'.

Additionally, brittle deformation in most fault gouges is observed to occur in a localized way, with displacement focused on a series of through-going and anastomosing shear surfaces (Craw and Upton, 2014). In polyphase gouges, graphite is quite commonly focused into these anastomosing shear surfaces (e.g. Nakamura et al., 2015; Kirilova et al., in press). So, in any natural fault that may have experienced a shear strain that we measure from its total displacement and the thickness of the deforming zone, it is also likely that individual graphite-bearing layers actually accommodated much higher shear strains, probably similar to the ones in our experiments. Therefore, correlation between bulk shear strain and R2 may actually be quite a good approximation because it is applicable to complex natural systems.

Reply to Response:

(a)  It is unclear to me, on what base the relationship between bulk (sample) strain and strain within the thin, deforming layer(s) should be linear over the range of all experiments. Is the number and the thickness of these layers known and if so, are they always constant? Otherwise, to my opinion a linear relation appears to be speculative.

(b) The complexity of a naturally deformed, polymineralic rock where graphite may form layers (which may tend to localize strain, which may be arranged in an anastomosing network and which may differ in thickness and hence experience a large variation in strain) is not necessarily identical to the situation in the experiments. Simply because an experiment is a complex, it's complexity may not automatically balance the complexities found in nature. Certainly there's a good chance that deformation localizes into graphite layers in nature, but it remains unclear to me why this should occur at the same (linear? as proposed above) ratio with respect to bulk strain as in the pure graphite experiments. It remains unclear to me why it should be likely that in a polymineralic rock, strain partition would between graphite and e.g. quartz in a similar way as in a pure graphite system.

Reviewer comment:

c) The authors use a mixture of surface related measures (friction coefficients, normal stress, slip rate) and volume measures (strain) which might be confusing in places, e.g. it might not be directly evident that the experiments with the 25 MPa normal stress should actually be stronger that the 5 MPa experiments.

Response:

With our mechanical data, presented in Figure 1, we are using standard measure- ments, i.e. double direct shear experiments on powdered material, to test the fric- tional properties of graphite. The vast literature (Brace & Byerlee, 1966; Byerlee 1978; Blampied et al., 1995; Marone, 1998; etc.) that put the base for fault strength is based on experiments similar to those presented in our work. Our experiments are performed at different values of normal stress (5 and 25 MPa) and at different sliding velocities (1-10-100 microns/s), and represent the deformation of a shear zone that has an initial thickness of 3 mm. In all the experiments with increasing displacement, i.e. with increasing strain, we observe an initial phase of strengthening until a peak stress is reached, then we observe strain weakening until a steady state friction is achieved. Mechanical data clearly show that: 1) Friction coefficient is lower at high normal stress and this means that the experimental fault at 25 MPa is weaker than the one at 5 MPa; 2) The sliding velocity does not influence too much the frictional properties of the ex- perimental fault. During deformation, a typical fabric develops within the entire shear zone (e.g. Marone, 1998, fig 11) and the fault strength is strongly related to fabric evolution. In other words, frictional strength, fabric and strain are strongly connected.

Reply to Response:

The answer of the authors reflects that a mixture of surface properties and strain is indeed confusing. Frition coefficients give a ratio of shear to normal stress but are not a measure of the actual strength of a volume of rock. Expressing sample "strength" in terms of friction coefficient may be more appropriate if it is assumed that the sample is actually deforming only along a  surface, however in that case the concept of strain is not existent.

Besides all, for the main and important part of the manuscript, I think the mechanical data (displacement-friction coefficient or strain-shear stress) are only relevant to make any point that the samples are deforming by a brittle, semi-brittle or ductile mechanism. The measures of strain (and in this case, estimates of strain inside the deforming layers of the sample) are those relevant to the points addressed in (2), the relation between Raman spectra and strain.

Reviewer comment:

2) R2 correlation with shear strain a) Following the number in formula 1 and as shown in Figure 3., shear shear is evaluated as a function of R2, the inverse would be logical and a fit is numerically not equivalent. Additionally, formula 1 is wrong.

> Response:
>
> We plotted R2 as a function of shear strain and fitted the curve one more time. On the attached figure (Fig. 1) you can see the resulting figure together with the new formula, which was automatically calculated by Excel. Could you please advise us if the revised version of the figure is correct?

Reviewer comment:

b) What is the physical basis that R2 and strain should have a power law relationship?

> Response:
>
> If we take the R2 value associated with fitting of a curve in Excel as a measure, then a power law function provides the best fit to the relationship between shear strain and R2. It may be possible to fit the curve using other types of equations, and we are open to your suggestions on what other functions we should try.

>> Reply to Response:
>>
>> The way data is evaluated (R2 as a function of "strain" and not the other way around) is now correct. However, why should it have now a quadratic relationship. At high shear strain, R2 will decrease again (also visible in your new plot)!  So the way to evaluate the data (R2=f(strain)) is correct, a quadratic fit however does not really make sense to me.

Reviewer comment:

3) Origin of D1,2 bands Obtaining Raman spectra at surface ledges on (001) or generally grain boundary regions, the appearance of the D bands has been observed (e.g.Tuinstra & Koenig, 1970; Pimenta et al. 2007). For small grain sizes, the ratio of G and D bands is actually used to establish a grain size determination. The authors mention that the increasing area of D1, D2 peaks is related to a decreasing crystallinity of graphite, is it possible that the crystallinity does not change but rather with a smaller grain size more grain boundary area with a disturbed lattice is measured? The authors estimate the minimum resolution of their optical system at 0.4 _m, however, from the text it becomes not clear whether this relates to the analysed point and/or if the analysed point could actually be identified and if so, whether measurements with a large R2 come from areas with a smaller grain size? If crystallinity is defined in relation to in-tragranular defect density/lattice perfection, it should be verified that only intragranular measurements are evaluated. If the grainsize is so small that most likely grain ag- gregates are measured, maybe the grain size effect could be corrected. The authors mention that they actually examine the structural disordering of graphite, so depending how this is defined, it needs to be considered separately to grain boundary effects.

> Response:
>
> We acknowledge that the measured increase in D bands can result both from (1) de- crease in crystallinity and (2) spectra obtained on grain boundaries. Furthermore, more surfaces are likely to be created as a result of brittle deformation. In addition, in natural fault zones graphite commonly appears with significantly smaller grain size (e.g. <1 micron in the Alpine Fault cataclasites; Kirilova et al, in press) than in our experimental samples. Thus, the calibrated Raman thermometer could yield temperatures that significantly underestimate the peak metamorphic temperatures experienced by the host rocks. Nevertheless, our experimental study proves that the calibrated Raman thermometers are unreliable in active tectonic settings.
>
> However, in our study: (1) We attempted to avoid grain boundaries as much as pos-sible, and thus most (if not all) of our measurements were obtained from intragranular areas; (2) our SEM data (fig. 4b) shows that the accumulated shear surfaces are compiled of grains > 10 microns in size, which is significantly larger than the laser spot size (approximately 412 nm). Therefore, we believe that the detected increase of D bands in our experimental data in fact reflects disorder of the internal structure of graphite rather than grain size reduction.

Reply to Response:

Both, a strongly reduced grain size as well as a reduced crystallinity will certainly lead to increased D bands and an underestimation of a RSCM temperature; this is the important message in the manuscript. However, based on the presented data, I do not clearly see how the two possible effects could be separated. It was mentioned in the manuscript that the intent was to avoid grain boundaries, however, to me it is not clear how this was done. Inspecting e.g. Figures 4a, a BSE image, I find it difficult to detect grains or grain boundaries. Are grain boundaries so much more obvious is the LM images of the Raman system? Is the assumed grain size of 10-50 µm really present in the thin deforming layers which were actually measured. A nice micrograph showing these features would be quite helpful.

Reviewer comment:

4) Applicability of a strain corrected RSCM thermometer a) in real rocks, it might be difficult to estimate strain and it might not be clear in which way deformation partitions between graphite and other minerals and so I'd encourage the authors to share some thoughts on how a strain correction should be applicable in a real-rock situation.

Response:

We agree that estimating shear strain in real rocks could be a challenging task. Therefore, the suggested calibration could be used only in the case when shear strain can be undoubtedly identified. Furthermore, in the initial version of the manuscript we acknowledge the importance of sliding velocities, and thus suggest that shear strain calibration may not be sufficient for reliable temperature estimates in active tectonic settings (L272-273).

Reply to Response:

a) It is often challenging to derive a reliable strain measure from a natural rock (even when neglecting e.g. a complex deformation-uplift history). Following the earlier arguments, I think it is even more challenging to estimate the strain that an individual graphite seam inside such a rock may have experienced. Assuming that the authors would establish their "strain-R2"-relation based on an estimate of the strain in the thin deforming layer, it would still be very difficult in natural cases to "undoubtedly identify" strain accumulated in the graphite seams. I think this is a sufficiently complex problem without actually having to start thinking about any effect of deformation rates (not sliding rates) or other parameters.

Reviewer comment:

b) A correction for the thermometer might depend on the relation of temperature and strain: e.g. during exhumation of a rock deformation takes place at increasingly lower temperature. Using a simple "strain" correction, would imply that the deformation temperatures during exhumation would not have to be considered. Given that most minerals show different deformation mechanisms at different temperatures, it might be reasonable to assume that this is the case as well for graphite. So strictly speaking, a correction should only consider lattice defects introduced by the identical process which is occurring in the calibration experiments. I'd encourage the authors to comment on this complication of such an effort.

Response:

As you have mentioned during exhumation of a rock, deformation takes place at increasingly lower temperature. However, graphite structure is sensitive only to increase in temperature i.e. graphite crystallinity increases with increasing temperature. Retrograde metamorphism is known not to affect the degree of graphite crystallinity that has been previously achieved. Thus, in the suggested scenario the accumulated shear strain will be the main parameter affecting graphite structural order. On the contrary, we expect that shear strain in high temperature conditions would yield significantly different results than the ones presented in our study.

Reply to Response:

b) This comment was not about the effect of retrograde metamorphism but rather on the effect of exhumation at increasingly lower temperatures, where it is not well known when brittle behaviour of graphite starts to dominant. The presented experiments nicely demonstrate that graphite deformed at room temperature shows a reduced RSCM temperature. Despite graphite deformation at higher temperature might indeed be a different story, it is primarily very challenging to identify the part of deformation accumulated in a graphite layer which was established under brittle conditions in order to apply the "strain-correction" derived by the presented, supposedly brittle experiments. I can perfectly imagine that higher temperature deformation does a similar RSCM temperature resetting to graphite, but there might be a different functional relation compared to the one which is intended in the present study.

Reviewer comment:

c) Using any thermometer to determine peak temperatures, often the measurements yielding the highest results are considered as representing peak conditions to overcome the problem of a partial lower temperature overprint. Given that some measurements in deformed samples still yield a low R2, it would be helpful to see where those measurements are actually determined. Are those from within grains while those with large D and D' bands contain areas with a high grain boundary density?

Response:

We attempted to obtain all our measurements within grains. Therefore, we do not expect to be observing an effect from a variable grain boundary density.

Reply to Response:

Similarly to my comments above, a convincing micrograph may enhance the manuscript. The Raman microscope seems to have a normal optical system, so a light micrograph where grains/boundaries in an analysed area would be recognisable could be a good addition to the manuscript. The authors state (L200ff) that it might have been possible that some undeformed grains might haven been accidentally measured through fractures in the surface. A nice micrograph might be able to explain why fractures could accidentally be overlooked but grain boundaries be clearly identified.

I think the discussion about the applicability of the RSCM thermometer correction and the actual strain dependency should be conducted very carefully. An estimate on the strain within the deforming layer (e.g. via the thickness of the deforming layer, providing an upper estimate if it is assumed that all/most deformation is realised within those layers) in comparison to the minimum strain estimate derived from the bulk sample (if strain is correctly derived) may provide a more extensive explanation under which assumptions a "strain-correction" could realistically be applied.

All the best, Rüdiger

---

## Referee Comment (RC3) · O. Kiyokazu (Referee) · 22 Sep 2017

This manuscript describes the effect of shearing on crystallinity of graphite by conducting biaxial friction experiments for powder of highly-crystalized graphite and micro-Raman analysis of the recovered sample. The manuscript documents systematic increase of crystallinity index, R2 value, which is widely used to determine paleo-maximum temperature of metamorphic rocks, with increasing applied shear strain. Recently, much attention has been given to carbonaceous materials (including graphite) in and around a fault zone because of its utilities as a heat anomaly detector, displacement indicator, and lubricating agent of fault. However, crystallographic changes of graphite, especially in relation to the fault activities, are not well understood except for the effect of temperature alone. The objective of the manuscript is straightforward and

their results seem sound. However, I found that some additional information and data are required and also there is room for further discussion and improvement. Hence I recommend accepting it with minor revisions.

General comments (1) The discussion about R2 and "shear strain" The main conclusion of this manuscript is R2 value of graphite increases (which implies decrease of crystallinity) with increasing applied "shear strain". However, "shear strain" you calculated corresponds to "bulk shear strain", and the bulk shear strain and microscopic shear strain which exactly concentrated into the narrow slip zone is quite different. Degree of compaction may differ depends on normal stress applied, hence the "bulk shear strain" reflects not only shearing but also compaction. Also I would suspect thinning due to leakage of the gouge took place under 25 MPa experiments, especially for slip rate of 100 $\mu$m/s (Exp.10). All these issues make it difficult to extract an effect of shearing on the increments of R2 value. To solve this problem, I would suggest using total frictional work (shear stress*displacement) in addition to shear strain, and to discuss its relationship to R2 value.

(2) The relationship between R2 value and graphite "crystallnity" As another referee also mentioned, D bands (and R2 value) reflect amount of grain boundary (edge of grapheme sheet) in addition to intracrystalline defects, so determining which process is dominant in your setting becomes another problem to be addressed. I could faintly see very small platelets of graphite (< 1$\mu$m) in your photograph (Figure 4e), but damages during shin-section making also make this kind of roughness. I think it is better to provide high-resolution SEM images of the slip surface if you could not make good thin section. On the other hand, I think you should mention that the friction still remains low and stable if you applied shear strain >40. This feature may suggest the graphite on the slip surface still maintain its crystal perfection. In that sense, increments of R2 value of sheared graphite attributable mainly to grain size reduction but not amorphization.

Specific comments

Line 46; I would happy if you add Oohashi et al. (2013, JGR) in the reference.

Line 60; I think initial thickness of gouge layer varies depends on applied normal stress if you put same weight of graphite powder for each experiments (becomes thinner under high normal stress). Did you change amount of graphite for normal stress of 5 MPa and 25 MPa to ensure to form exactly the same 3-mm thickness? This question is arises from why large shear strain is calculated from the experiments under 25 MPa normal stress (off course, I understand your explanation about compaction). Additionally, I would suspect thinning due to leakage of the gouge took place under 25 MPa experiments, especially for slip rate of 100 $\mu$m/s (Exp.10) because the gouge thickness seems to became less than half of the initial thickness.

Line 109-110; The authors documented $\mu$ss does not depend on slip rates, and it remains constant for all experiments. However, I see clear relationship between $\mu$ at d=14-20 mm and slip rates; $\mu$ decreases with decreasing slip rates for $\sigma$n=5 MPa, and $\mu$ decreases with increasing slip rates for $\sigma$n=25 MPa.

Line 141-143; The authors explain graphite crystalinity decreases with increasing slip rates for samples sheared under $\sigma$n=25 MPa, and no slip rates dependence is found for samples sheared under $\sigma$n=5 MPa. However, as you concluded, increase of R2 value can be attributed to applied shear strain but not to slip rates. I think you can not discuss direct relationship between R2 value and slip rates unless you conduct various slip-rates experiments at exactly the same shear strain.

Line 181; I would suggest referring Di Toro et al. (2011, Nature) instead of Nakatani (2001).

Table 2 and Figure 3 Please add errors and error bars for R2 value.

Hope this helps,

Kiyokazu Oohashi, Yamaguchi University, Japan

---

## Author Comment (AC2) · 28 Oct 2017

Oohashi-San,

Thank you for spending the time to leave an interactive comment in the open discussion of our manuscript. We acknowledge your expertise in experimental studies on graphite, and thus we are very pleased to receive a positive feedback from you. We consider your comments greatly beneficial for a successful revision of this manuscript, and thus we have taken account of your suggestions to provide more observational data in a revised version of the manuscript. Below we have copy-pasted and then responded to the comments you made in your review.

General comments

[Figure]

Reviewer's comment: (1) The discussion about R2 and "shear strain" The main conclusion of this manuscript is R2 value of graphite increases (which implies decrease of crystallinity) with increasing applied "shear strain". However, "shear strain" you calculated corresponds to "bulk shear strain", and the bulk shear strain and microscopic shear strain which exactly concentrated into the narrow slip zone is quite different. Degree of compaction may differ depends on normal stress applied, hence the "bulk shear strain" reflects not only shearing but also compaction. Also, I would suspect thinning due to leakage of the gouge took place under 25 MPa experiments, especially for slip rate of 100 $\mu$m/s (Exp.10). All these issues make it difficult to extract an effect of shearing on the increments of R2 value. To solve this problem, I would suggest using total frictional work (shear stress*displacement) in addition to shear strain, and to discuss its relationship to R2 value.

Response: This issue was also noted by another referee and we have documented our response to both comments in the other response to review. We agree that the measured bulk shear strain is most probably significantly lower than the shear strain accumulated only within the thin shear surfaces. However, we expect the shear strain variations within these surfaces to be linearly correlated with the measured bulk shear strain within a sample, but we recognise and acknowledge that we are only able to calculate a 'rough approximation' from our experimental data. In addition, we have now calculated frictional work as you suggested (Table 2 and Fig. 3b). Reviewer's comment: (2) The relationship between R2 value and graphite "crystallnity" As another referee also mentioned, D bands (and R2 value) reflect amount of grain boundary (edge of grapheme sheet) in addition to intracrystalline defects, so determining which process is dominant in your setting becomes another problem to be addressed.

Response: As we reported in a previous referee's response, we attempted to avoid grain boundaries as much as possible, which was easily done due to the fact that the laser spot size (412 nm) was much smaller than the graphite grains in our samples (>10 microns, fig 4b). We acknowledge that some of the spectra may have been affected by

an increase in the grain boundary density, however occasional measurements of this sort are unlikely to affect the average R2 per sample. Thus, we believe that the detected increase in D bands in our experimental data reflects disorder of the internal structure of graphite rather than grain size reduction. Nevertheless, a discussion addressing this topic is added in lines 225-233. Reviewer's comment: I could faintly see very small platelets of graphite (< $1\mu$m) in your photograph (Figure 4e), but damages during shin-section making also make this kind of roughness. I think it is better to provide high-resolution SEM images of the slip surface if you could not make good thin section.

Response: (1) We collected Raman spectra directly from the top of the sheared graphite gouges (referred to as shiny surfaces in the manuscript) to avoid damage induced during thin section making (the thin sections were made from other parts of the preserved experimental samples). High resolution SEM images of these surfaces (Fig. 4a and b) were also collected directly from the tops of the layers (labelled as XY sections in the manuscript). (2) We imaged the zone underlying the shiny surfaces by cutting thin sections perpendicular to them. Thus, the small platelets of graphite (< $1\mu$m) in Figure 4e, that you refer to, might have been affected by sample preparation. However, Raman spectra were not obtained from these samples, and thus the reported D bands do not reflect any damage induced during thin section making.

Reviewer's comment: On the other hand, I think you should mention that the friction still remains low and stable if you applied shear strain >40. This feature may suggest the graphite on the slip surface still maintain its crystal perfection. In that sense, increments of R2 value of sheared graphite attributable mainly to grain size reduction but not amorphization.

Response: Friction coefficients remain low and stable throughout all experiments. This is reflected in Fig. 1 and in lines 185-186. Our microstructural data clearly indicate partial structural disorder of the graphite structure, so we don't think that the reduction in the Raman spectra can be reasonably attributed just to grain size reduction. Also, please note there is no evidence of complete amorphization – graphite in all ex-
Interactive
comment

periments remains crystalline even though significant defects in the graphite structure were introduced.

Specific comments: Reviewer's comment: Line 46; I would happy if you add Oohashi et al. (2013, JGR) in the reference. Response: The citation has been added into the reference list (in lines 378-380). Thank you for reminding us to properly acknowledge this excellent previous study.

Reviewer's comment: Line 60; I think initial thickness of gouge layer varies depends on applied normal stress if you put same weight of graphite powder for each experiments (becomes thinner under high normal stress). Did you change amount of graphite for normal stress of 5 MPa and 25 MPa to ensure to form exactly the same 3-mm thickness? This question is arises from why large shear strain is calculated from the experiments under 25 MPa normal stress (off course, I understand your explanation about compaction). Additionally, I would suspect thinning due to leakage of the gouge took place under 25 MPa experiments, especially for slip rate of 100 $\mu$m/s (Exp.10) because the gouge thickness seems to became less than half of the initial thickness. Response: The mass of graphite was not changed throughout the various experiments and hence we acknowledge the effect of normal stresses on the attained shear strain in lines 190-192 (as you have also noted in your comment). Your concern about potential leakage of gouge material during Exp. 10 is based on the dramatic layer thinning recorded at the end of this experiment. However, figure 1b clearly shows a linear trend of increase in shear strain (based on layer thinning) with increase of sliding velocities in the experiments under normal stresses of 25 MPa (ranging from 21,45 through 31.86 to 46,77 in Exp. 8, 9 and 10 respectively). Therefore, we believe that leakage of gouge material is unlikely to have affected the measured layer thickness.

Reviewer's comment: Line 109-110; The authors documented $\mu$ss does not depend on slip rates, and it remains constant for all experiments. However, I see clear relationship between $\mu$ at d=14-20 mm and slip rates; $\mu$ decreases with decreasing slip rates for $\sigma$n=5 MPa, and $\mu$ decreases with increasing slip rates for $\sigma$n=25 MPa. Response:

Dependence of $\mu$ss on slip rates is not suggested by our data (Table 1). Instead we observe slight variations in 2 of the performed experiments (Exp. 4 and 10). This is reflected in the manuscript in lines 119-120.

Reviewer's comment: Line 141-143; The authors explain graphite crystalinity decreases with increasing slip rates for samples sheared under _n=25 MPa, and no slip rates dependence is found for samples sheared under _n=5 MPa. However, as you concluded, increase of R2 value can be attributed to applied shear strain but not to slip rates. I think you can not discuss direct relationship between R2 value and slip rates unless you conduct various slip-rates experiments at exactly the same shear strain. Response: In lines 151-153, we simply compare the observed decrease in graphite crystallinity with the conditions of the experiments but do not imply direct relationship between R2 and slip rates. Then (lines 153-154), we conclude that 'graphite appears as most disordered in the experiments where the highest shear strain was achieved'.

Reviewer's comment: Line 181; I would suggest referring Di Toro et al. (2011, Nature) instead of Nakatani (2001). Response: Thank you for the suggestion. The reference is now updated. (line 198)

Reviewer's comment: Table 2 and Figure 3 Please add errors and error bars for R2 value. Response: Error estimates are now added in table 2 and line 432.

---

## Author Response (AR1)

Dr. Renée Heilbronner

Handling Topical Editor, Solid Earth

Dear Renée Heilbronner,

We are pleased to submit a revised version of this manuscript, entitled 'Structural Disorder of Graphite and Implications for Graphite Thermometry' to you, for consideration for publication in Solid Earth.

Herein, we investigated the possibility of mechanical modifications of graphite structure in laboratory deformation experiments. We document systematic decrease in graphite crystallinity as a function of increasing shear strain during shear experiments with aseismic sliding velocities. Our results contradict the paradigm that the degree of graphite crystallinity is determined by irreversible maturation of carbonaceous material. This finding has implications for the calibrated graphite 'thermometer' as mechanically induced disorder of the graphite structure may lead to temperature calculations that significantly underestimate the peak metamorphic temperatures in deformed rocks. Therefore, we suggest that the current graphite 'thermometer' should be re-evaluated.

We have substantially revised the manuscript taking into account reviewer`s suggestions, as well as some ideas that have been discussed among co-authors in the months since first submission. In particular, we further examined the nature of the processes that operated during shearing and resulted in the documented structural disorder of graphite by implementing transmission electron microscopy. Also, we have added Toru Takeshita as a co-author because of his valuable contribution to the current manuscript.

We have copied the reviews below and addressed each comment in turn. Our responses are indicated in green text. We have also used track changes to document revisions to the manuscript and are resubmitting both a track changes, and changes accepted versions. We refer to revisions in the manuscript by line numbers – these are correct with respect to the revised version with changes accepted.

Thank you for your consideration of this manuscript.

Regards,

Martina Kirilova
Corresponding Author
martina.a.kirilova@gmail.com

On behalf of the authors: Virginia Toy, Jeremy S. Rooney, Carolina Giorgetti, Keith C. Gordon, Cristiano Collettini and Toru Takeshita.
In the manuscript "Structural Disorder of Graphite and Implications for Graphite Thermometry" s2-2017-74, Kirilova et al. report the results of Raman spectroscopy on an experimentally sheared, synthetic graphite gouge. Raman spectroscopy on carbonaceous material (RSCM) has become a frequently used geothermometer assuming that the crystallinity of graphite represents peak metamorphic conditions. In contrast to geothermometers based on mineral compositions where a retrograde overprint can often be easily recognized, preservation of peak metamorphic conditions under deformation might not be easily recognised in graphite where no compositon changes are to be expected. Accordingly, this contribution is of special interest to researchers, either using RSCM or having to evaluate RSCM derived data. Testing the RSCM thermometer against experimentally deformed graphite gouge is novel and the results of the presented study are very interesting and strongly suggest that the applicability RSCM thermometry to deformed rocks - and accordingly to almost all metamorphic rocks - needs a careful evaluation. This ms highlights that not all parameters to unequivocally interpret RSCM are sufficiently understood. The authors of the ms suggest a correction of the RSCM thermometer based on shear strain, related to their observations in the synthetic graphite gouge.

The manuscript is concise, easy to read and provides a valuable insight into a neglected problem of RSCM. However, there are points that are either not entirely consistent, that need further clarification or corrections, especially in the obtained relation to sample strain and the determination of crystallinity and those should be addressed (listed below). Overall I recommend that the manuscript is suitable for publication after moderate revisions.

Following comments should be addressed.

1) Relation to strain/stress
a) A "shear strain" is calculated by summing up the ratios of displacement increments and thickness. As it appears the samples are thinning with increasing displacement, this "shear strain" is neglecting the thinning component, the derived "shear strain" cannot be used to calculate a strain ellipse (no functional relation) and overestimates strain in the sample. A more correct procedure would be one where progressive simple and pure shear are treated to occur concurrently, and reporting a unique measure of strain.

Response:
By "measured layer thickness" we mean the layer thickness measured at the same record number at which the shear displacement increments were measured, i.e. it is the instantaneous layer thickness. Consequently, we take into account the sample thinning with increasing displacement. Perhaps we need to better explain this in the manuscript.

However, we recognize that you have recently recalculated strains in simple shear experiments that take the layer thinning into account (Kilian, R., & Heilbronner, R. 2017, April. Texture transition in experimentally deformed quartzite. In EGU General Assembly Conference

Abstracts, Vol. 19, p. 6966). We hope you can advise us in more detail how we could most appropriately make use of such calculations in our current study.

Reply to Response:

Summing up individual steps (delta displacement/current thickness) does not produce a shear strain which accounts for the deformation imposed on the sample. This measure could be termed "apparent shear strain" but it overestimates the true (shear) strain. This apparent shear strain cannot be used to derive a strain ellipse (and accordingly strain related measures i.e. LINFS, ISA, ... any measure of strain). Any correlations of e.g. microstructural features or any other supposedly strain dependent material property should greatly benefit from a correct measure of strain. For a simple procedure to derive bulk strain under the assumption of simultaneous thinning and shearing see e.g. Fossen&Tikoff (1993).

Response 2: Thank you for the additional explanation and the reference provided. In support to our shear strain estimates we have also calculated total frictional work as suggested by Reviewer 2 to avoid this issue.

b) The authors report (and obtain their measurements) from the shiny surface where they assume along which most of the displacement is realised. Assuming that this surface is actually a thin layer, the strain within that layer must always be larger than the strain derived from the entire sample. Hence one might speculate that a functional relation between the determined "shear strain" and R2 is at best a rough approximation. In case, as indicated by some of the comments on the microstructure, the compaction (sample thinning?) is localised as well and not homogeneous, a bulk sample, "strain estimate" is even more likely to be unrelated to the state of deformation in the analysed layer.

Response:

We agree with your statement that the measured bulk shear strain is most probably significantly lower than the shear strain accumulated only within the thin shear surfaces. However, we expect the shear strain variations within these surfaces to be linearly correlated with the measured bulk shear strain within a sample. Thus, we believe that a correlation between shear strain and R2 is in fact possible. Nevertheless, we will relate to this relationship as a 'rough approximation'.

Additionally, brittle deformation in most fault gouges is observed to occur in a localized way, with displacement focused on a series of through-going and anastomosing shear surfaces (Craw and Upton, 2014). In polyphase gouges, graphite is quite commonly focused into these anastomosing shear surfaces (e.g. Nakamura et al., 2015; Kirilova et al., in press). So, in any natural fault that may have experienced a shear strain that we measure from its total displacement and the thickness of the deforming zone, it is also likely that individual graphite-bearing layers actually accommodated much higher shear strains, probably similar to the ones in our experiments. Therefore, correlation between bulk shear strain and R2 may actually be quite a good approximation because it is applicable to complex natural systems.

Reply to Response:

(a) It is unclear to me, on what base the relationship between bulk (sample) strain and strain within the thin, deforming layer(s) should be linear over the range of all experiments. Is the number and the thickness of these layers known and if so, are they always constant? Otherwise, to my opinion a linear relation appears to be speculative.

(b) The complexity of a naturally deformed, polymineralic rock where graphite may form layers (which may tend to localize strain, which may be arranged in an anastomosing network and which may differ in thickness and hence experience a large variation in strain) is not necessarily identical to the situation in the experiments. Simply because an experiment is a complex, it's complexity may not automatically balance the complexities found in nature. Certainly, there's a good chance that deformation localizes into graphite layers in nature, but it remains unclear to me why this should occur at the same (linear? as proposed above) ratio with respect to bulk strain as in the pure graphite experiments. It remains unclear to me why it should be likely that in a polymineralic rock, strain partition would between graphite and e.g. quartz in a similar way as in a pure graphite system.

Response 2: In our experimental samples these layers tend to form at systematic spacing controlled by the spacing of teeth on the forcing blocks. However, we acknowledge that the strain accommodated within each shiny surface will be much higher than the bulk strain accommodated by the gouge layer. This is now acknowledged in lines 214-216. Furthermore, polyphase natural gouges that comprise anastomosing networks of localised shear zones between more rigid clasts will also contain graphite layers that accommodate greater strains individually than the bulk strain within the layer. However, we recommend that the relationship we have determined can only be directly applied to natural cases if the spacing and thickness of localised layers compared to total layer thickness is similar to that of our experiments.

c) The authors use a mixture of surface related measures (friction coefficients, normal stress, slip rate) and volume measures (strain) which might be confusing in places, e.g. it might not be directly evident that the experiments with the 25 MPa normal stress should actually be stronger that the 5 MPa experiments.

Response:
With our mechanical data, presented in Figure 1, we are using standard measurements, i.e. double direct shear experiments on powdered material, to test the frictional properties of graphite. The vast literature (Brace & Byerlee, 1966; Byerlee 1978; Blampied et al., 1995; Marone, 1998; etc.) that put the base for fault strength is based on experiments similar to those presented in our work. Our experiments are performed at different values of normal stress (5 and 25 MPa) and at different sliding velocities (1-10-100 microns/s), and represent the deformation of a shear zone that has an initial thickness of 3 mm. In all the experiments with increasing displacement, i.e. with increasing strain, we observe an initial phase of strengthening until a peak stress is reached, then we observe strain weakening until a steady state friction is achieved. Mechanical data clearly show that: 1) Friction coefficient is lower at high normal stress and this means that the experimental fault at 25 MPa is weaker than the one at 5 MPa; 2) The sliding velocity does not influence too much the frictional properties of the experimental fault. During deformation, a typical fabric develops within the entire shear zone (e.g. Marone, 1998, fig 11) and the fault strength is strongly related to fabric evolution. In other words, frictional strength, fabric and strain are strongly connected.

Reply to Response:

The answer of the authors reflects that a mixture of surface properties and strain is indeed confusing. Frition coefficients give a ratio of shear to normal stress but are not a measure of the actual strength of a volume of rock. Expressing sample "strength" in terms of friction coefficient may be more appropriate if it is assumed that the sample is actually deforming only along a surface, however in that case the concept of strain is not existent.

Besides all, for the main and important part of the manuscript, I think the mechanical data (displacement-friction coefficient or strain-shear stress) are only relevant to make any point that the samples are deforming by a brittle, semi-brittle or ductile mechanism. The measures of strain (and in this case, estimates of strain inside the deforming layers of the sample) are those relevant to the points addressed in (2), the relation between Raman spectra and strain.

Response 2: We agree with your comment.

2) R2 correlation with shear strain
a) Following the number in formula 1 and as shown in Figure 3., shear shear is evaluated as a function of R2, the inverse would be logical and a fit is numerically not equivalent. Additionally, formula 1 is wrong.

Response:
We plotted R2 as a function of shear strain and fitted the curve one more time. On Figure 3_version 2 (attached) you can see the resulting figure together with the new formula, which was automatically calculated by Excel. Could you please advise us if the revised version of the figure is correct?

[Figure]

b) What is the physical basis that R2 and strain should have a power law relationship?

Response:

If we take the R2 value associated with fitting of a curve in Excel as a measure, then a power law function provides the best fit to the relationship between shear strain and R2. It may be possible to fit the curve using other types of equations, and we are open to your suggestions on what other functions we should try.

Reply to Response:

The way data is evaluated (R2 as a function of "strain" and not the other way around) is now correct. However, why should it have now a quadratic relationship. At high shear strain, R2 will decrease again (also visible in your new plot)! So the way to evaluate the data (R2=f(strain)) is correct, a quadratic fit however does not really make sense to me.

Response 2: Thank you for this comment. We plotted the figure by using Matlab instead of Excel, and we achieved much better results. This relationship is now described by a power function.

[Figure]

3) Origin of D1,2 bands Obtaining Raman spectra at surface ledges on (001) or generally grain boundary regions, the appearance of the D bands has been observed (e.g.Tuinstra & Koenig, 1970; Pimenta et al. 2007). For small grain sizes, the ratio of G and D bands is actually used to establish a grain size determination. The authors mention that the increasing area of D1, D2 peaks is related to a decreasing crystallinity of graphite, is it possible that the crystallinity does not change but rather with a smaller grain size more grain boundary area with a disturbed lattice is measured? The authors estimate the minimum resolution of their optical system at 0.4 _m, however, from the text it becomes not clear whether this relates to the analysed point and/or if the analysed point could actually be identified and if so, whether measurements with a large R2 come from areas with a smaller grain size? If crystallinity is defined in relation to intragranular defect density/lattice perfection, it should be verified that only intragranular measurements are evaluated. If the grainsize is so small that most likely grain aggregates are measured, maybe the grain size effect could be corrected. The authors mention that they actually examine the structural disordering of graphite, so depending how this is defined, it needs to be considered separately to grain boundary effects.

Response:

We acknowledge that the measured increase in D bands can result both from (1) decrease in crystallinity and (2) spectra obtained on grain boundaries. Furthermore, more surfaces are likely to be created as a result of brittle deformation. In addition, in natural fault zones graphite commonly appears with significantly smaller grain size (e.g. <1 micron in the Alpine Fault cataclasites; Kirilova et al, in press) than in our experimental samples. Thus, the calibrated Raman thermometer could yield temperatures that significantly underestimate the peak metamorphic temperatures experienced by the host rocks. Nevertheless, our experimental study proves that the calibrated Raman thermometers are unreliable in active tectonic settings.

However, in our study: (1) We attempted to avoid grain boundaries as much as possible, and thus most (if not all) of our measurements were obtained from intragranular areas; (2) our SEM data (fig. 4b) shows that the accumulated shear surfaces are compiled of grains > 10 microns in size, which is significantly larger than the laser spot size (approximately 412 nm). Therefore, we believe that the detected increase of D bands in our experimental data in fact reflects disorder of the internal structure of graphite rather than grain size reduction.

Reply to Response:

Both, a strongly reduced grain size as well as a reduced crystallinity will certainly lead to increased D bands and an underestimation of a RSCM temperature; this is the important message in the manuscript. However, based on the presented data, I do not clearly see how the two possible effects could be separated. It was mentioned in the manuscript that the intent was to avoid grain boundaries, however, to me it is not clear how this was done. Inspecting e.g. Figures 4a, a BSE image, I find it difficult to detect grains or grain boundaries. Are grain boundaries so much more obvious is the LM images of the Raman system? Is the assumed grain size of 10-50 μm really present in the thin deforming layers which were actually measured. A nice micrograph showing these features would be quite helpful.

Response 2: We show the size of grains comprising the shiny surface clearly in Figure 4b. Furthermore, the overall crystallinity of a sample will naturally reduce with grain size reduction since that increases the grain surface (volume ratio and grain surfaces are much less crystalline than their interiors).

4) Applicability of a strain corrected RSCM thermometer
a) in real rocks, it might be difficult to estimate strain and it might not be clear in which way deformation partitions between graphite and other minerals and so I'd encourage the authors to share some thoughts on how a strain correction should be applicable in a real-rock situation.

Response:

We agree that estimating shear strain in real rocks could be a challenging task. Therefore, the suggested calibration could be used only in the case when shear strain can be undoubtedly identified. Furthermore, in the initial version of the manuscript we acknowledge the importance of sliding velocities, and thus suggest that shear strain calibration may not be sufficient for reliable temperature estimates in active tectonic settings (lines 304-308).

Reply to Response:

a) It is often challenging to derive a reliable strain measure from a natural rock (even when neglecting e.g. a complex deformation-uplift history). Following the earlier arguments, I think it is even more challenging to estimate the strain that an individual graphite seam inside such a rock may have experienced. Assuming that the authors would establish their "strain-R2"-relation based on an estimate of the strain in the thin deforming layer, it would still be very difficult in natural cases to "undoubtedly identify" strain accumulated in the graphite seams. I think this is a sufficiently complex problem without actually having to start thinking about any effect of deformation rates (not sliding rates) or other parameters.

Response 2: We are glad you agree.

b) A correction for the thermometer might depend on the relation of temperature and strain: e.g. during exhumation of a rock deformation takes place at increasingly lower temperature. Using a simple "strain" correction, would imply that the deformation temperatures during exhumation would not have to be considered. Given that most minerals show different deformation mechanisms at different temperatures, it might be reasonable to assume that this is the case as well for graphite. So strictly speaking, a correction should only consider lattice defects introduced by the identical process which is occurring in the calibration experiments. I'd encourage the authors to comment on this complication of such an effort.

Response: As you have mentioned during exhumation of a rock, deformation takes place at increasingly lower temperature. However, graphite structure is sensitive only to increase in temperature i.e. graphite crystallinity increases with increasing temperature. Retrograde metamorphism is known not to affect the degree of graphite crystallinity that has been previously achieved. Thus, in the suggested scenario the accumulated shear strain will be the main parameter affecting graphite structural order. On the contrary, we expect that shear strain in high temperature conditions would yield significantly different results than the ones presented in our study.

Reply to Response:

b) This comment was not about the effect of retrograde metamorphism but rather on the effect of exhumation at increasingly lower temperatures, where it is not well known when brittle behaviour of graphite starts to dominant. The presented experiments nicely demonstrate that graphite deformed at room temperature shows a reduced RSCM temperature. Despite graphite deformation at higher temperature might indeed be a different story, it is primarily very challenging to identify the part of deformation accumulated in a graphite layer which was established under brittle conditions in order to apply the "strain-correction" derived by the presented, supposedly brittle experiments. I can perfectly imagine that higher temperature deformation does a similar RSCM temperature resetting to graphite, but there might be a different functional relation compared to the one which is intended in the present study.

Response 2: True. We did already acknowledge that ductile as well as brittle mechanisms may impact the thermometric calibration, but we don't have data to address that in the current study (lines 275-279).

c) Using any thermometer to determine peak temperatures, often the measurements yielding the highest results are considered as representing peak conditions to overcome the problem of a partial lower temperature overprint. Given that some measurements in deformed samples still yield a low R2, it would be helpful to see where those measurements are actually determined. Are those from within grains while those with large D and D' bands contain areas with a high grain boundary density?

Response:
We attempted to obtain all our measurements within grains. Therefore, we do not expect to be observing an effect from a variable grain boundary density.

Reply to Response:

Similarly to my comments above, a convincing micrograph may enhance the manuscript. The Raman microscope seems to have a normal optical system, so a light micrograph where grains/boundaries in an analysed area would be recognisable could be a good addition to the manuscript. The authors state (L200ff) that it might have been possible that some undeformed grains might haven been accidentally measured through fractures in the surface. A nice micrograph might be able to explain why fractures could accidentally be overlooked but grain boundaries be clearly identified.

I think the discussion about the applicability of the RSCM thermometer correction and the actual strain dependency should be conducted very carefully. An estimate on the strain within the deforming layer (e.g. via the thickness of the deforming layer, providing an upper estimate if it is assumed that all/most deformation is realised within those layers) in comparison to the minimum strain estimate derived from the bulk sample (if strain is correctly derived) may provide a more extensive explanation under which assumptions a "strain-correction" could realistically be applied.

Response 2: Here we provide a screen capture as an example of the areas we have been measuring with the Raman microspectrometer.

[Figure]

The manuscript and the reader would benefit from some definitions: crystallinity (vs structural ordering for example), interpretation of the peaks/bands in the Raman spectrum, (G being sp2 activation, D1 most likely to intraplane defects and D2 to out-of plane defects, sp3 defects ...) should be introduced and defined. Why are sliding velocity, slip rate, shear velocity are all used synonymously? The figures regarding the microstructural description might benefit from a better resolution since some feature referred to in the text cannot be seen in the figures (comments below).

Minor comments on the manuscript (l=line):
l 41:friction coefficient
Corrected. Now in line 43.

l 52: no "or"
Removed. Now in line 55.

l 54:160 m are maximum grain size?
Yes, 160 microns is the maximum grain size. This is now clarified in the text in line 57.

l 54: annealed instead of "cooked"?
Replaced. Now in line 57

l 85: Please introduce G,D1,D2
The information has been added in lines 89-90.

l 102: coefficient (please also correct that in the figures if you prefer to stay with friction coefficients instead of shear stress)
Thanks for the comment. This has been accordingly corrected throughout the manuscript.

l 103: coefficient
Added. Now in line 113.

l107: Plots ofat all slip rates ..: I do not see the gradual decrease of peak (numbers, if I'm not mistaken are 0.43, 0.43, 0.41, so I wouldn't call this a gradual decrease)
Thank you for the comment. The text is now modified accordingly (lines 119-120 ).

l 108 slip rates...shear velocity: is there any difference? Elsewhere like in the tables there is slip rate,sliding velocity, why are they all used synonymously, please settle for one term.
The text has been modified. We now use 'sliding velocities' throughout the manuscript.

l 110:Where are values of $\mu_{ss}$ are read off?
$\mu_{ss}$ were read at the end of each experiment. This information is now added to the manuscript in line 119.

l 135: retained instead of produced
Replaced with 'collected'. Now in line 145.

l 137: varies
Corrected. Now in line 147.

l140: high-pressure experiments: should be high normal stress experiments
Corrected. Now in line 150.

l 152: fractures with random orientation compared to the slip direction: I can't see those
These fractures are cross-cutting the documented slikenslides. See figure 4a.

l 155:...well-compacted layer
Corrected.

Fig 4c: hard to see in the Figure what is described in the text
The figure has been modified.

l156: "randomly oriented ... Fig 4d": Is that actually confirmed or just based on visual impression
We interpret these from our microstructural observations.

l 158: "weak fabric development (Fig. 4e)" I can't see this in the figure
We refer to the layer comprised of aligned graphite grains. On the figure it is marked with an arrow, labelled as 'foliation'.

l 159:"filled with smaller graphite grains...Fig 4f": also here, it is hard to see, I hope a better resolution of the image and some arrows may help
The figure has been modified.

l 176: "more efficient reorientation"What should that be, rotation per time, alignment after strain...?
We refer to previous interpretations by Morrow et al. (2000) that attribute the initial $\mu$peak to the work involved in rotating the graphite grains with their (001) planes sub-parallel to the shear surfaces. This concept is introduced in the previous paragraph in lines 187-192.

l 178: This clear trend is not so clear to me.
The text has been modified accordingly.

l 180: "partial frictional heating": What should that be?
'partial' has been removed from the text for better clarity (line 197). Oohashi et al (2011) documented graphitization of carbonaceous material during high-velocity shear experiments due to frictional heating.

l 183 ff:The relation of compaction (volume change), thinning and the apparent dependency of"shear strain" on normal stress and displacement rate should be reconsidered, given that no suitable measure for strain is used
We have previously justified our estimates of shear strain and we have additionally calculated total frictional work as it was suggested by reviewer 2. We hope this revised version of the manuscript will satisfy the reviewer.

l 188: replace high-pressure with high normal stress
Replaced. Now in line 205.

l 207: "stable mineral" from the thermodynamic point of view it is the stable mineral, so please explain your definition of "stable".
By 'stable' we mean that graphite crystallinity is known to increase with increase in temperature, and to remain unaffected by retrograde metamorphism (Buseck and Beyssac, 2014). This concept was introduced in the introduction in lines 27-30. For better clarity a reference is now added to the text in line 236.

l 291-223: There is a mixture of lithostatic pressure and normal stress: Any effect that is observed at higher normal stress does not mean there is a "significant effect of lithostatic pressure". Normal stress is not equivalent to confining/lithostatic pressure.
We now say '… it can be exposed to different lithostatic pressures, and hence different normal stresses ...' (lines 249) because normal stress (both in nature or experiments) results from a lithostatic pressure and a differential stress.

l 234: "... microstructural observations provide some indications of the deformation processes..." These observations could be enhanced!
We have enhanced our discussion of microstructural observations. In addition, TEM analyses were also performed. (lines 264-274)

l 239: "plastic mechanisms" Such as?
We refer to any mechanisms that are likely to operate at higher temperatures and confining pressures than the ones implemented during our experiments.

l 240: "plastic deformation"
What should that be in contrast to the presented experiments where a gouge seems to flow by a rate independent mechanism?
We refer to deformation that takes place at higher temperatures and pressures than the ones implemented in our experiments (line 277).

l 244: "crystallographic structure ..." This seems to be a speculation neglecting a grain size effect.
We have clarified in the discussion that we collected our Raman spectra (mainly) from intragranular areas. Any occasional measurements along grain boundaries are unlikely to affect the overall analysis.

l 262ff: Any fit should consider the measure of strain as the independent variable, not as shown and calculated as the dependent one. Independent from that, (1) is not correct.

We have modified our plot and introduced new equation (line 302)

l 273: "We observe a trend..."So it appear there is an increasing thinning which however needs not to translate into an increased shear strain.

We take into account the sample thinning with increasing displacement and based on this relationship we calculate shear strain at equivalent record numbers.

l350: "approximate average crystallinity": What should that be?

We agree that 'approximate average crystallinity' is a bit confusing. We now say 'average crystallinity'(line 431).

Figures and Tables:
Table 1 and table 2 are mostly redundant and differ just by 1 column.

We acknowledge that the tables are very similar to each other. However, in table 2 the experiments are ordered in respect to increasing shear strain (rather than experiemental conditions) to emphasize the relationship between shear strain and R2.

Figure 1 It would be nice to use identical colors in (c) and (d) representing the displacement. Y-axis should read friction coefficient.

We have chosen to keep consistency with the plots in a and b instead. Meaning that exp. 1 and 5 have identical colours in all plots. Y-axis has been changed to 'friction coefficient'.

Figure 4 could strongly benefit from some images that more clearly show what is described in the text.

Thank you for the comment. We have modified the figure.

Supplementary material: If "int" stands for intensity and "pos" for wavenumber, it seems like the column headers are misplaced.

We are grateful for noticing this mistake. The supplementary material has been corrected.

Structural Disorder of Graphite and Implications for Graphite Thermometry
Martina Kirilova
2017-09-22
This manuscript describes the effect of shearing on crystallinity of graphite by conducting biaxial friction experiments for powder of highly-crystalized graphite and micro- Raman analysis of the recovered sample. The manuscript documents systematic increase of crystallinity index, R2 value, which is widely used to determine paleomaximum temperature of metamorphic rocks, with increasing applied shear strain. Recently, much attention has been given to carbonaceous materials (including graphite) in and around a fault zone because of its utilities as a heat anomaly detector, displacement indicator, and lubricating agent of fault. However, crystallographic changes of graphite, especially in relation to the fault activities, are not well understood except for the effect of temperature alone. The objective of the manuscript is straightforward and their results seem sound. However, I found that some additional information and data are required and also there is room for further discussion and improvement. Hence I recommend accepting it with minor revisions.

General comments
Reviewer`s comment:
(1) The discussion about R2 and "shear strain" The main conclusion of this manuscript is R2 value of graphite increases (which implies decrease of crystallinity) with increasing applied "shear strain". However, "shear strain" you calculated corresponds to "bulk shear strain", and the bulk shear strain and microscopic shear strain which exactly concentrated into the narrow slip zone is quite different. Degree of compaction may differ depends on normal stress applied, hence the "bulk shear strain" reflects not only shearing but also compaction. Also, I would suspect thinning due to leakage of the gouge took place under 25 MPa experiments, especially for slip rate of 100 µm/s (Exp.10). All these issues make it difficult to extract an effect of shearing on the increments of R2 value. To solve this problem, I would suggest using total frictional work (shear stress*displacement) in addition to shear strain, and to discuss its relationship to R2 value.

Response:

This issue was also noted by another referee and we have documented our response to both comments in the other response to review. We agree that the measured bulk shear strain is most probably significantly lower than the shear strain accumulated only within the thin shear surfaces. However, we expect the shear strain variations within these surfaces to be linearly correlated with the measured bulk shear strain within a sample, but we recognise and acknowledge that we are only able to calculate a 'rough approximation' from our experimental data. In addition, we have now calculated frictional work as you suggested (Table 2 and Fig. 3b).

Reviewer`s comment:
 (2) The relationship between R2 value and graphite "crystallnity" As another referee also mentioned, D bands (and R2 value) reflect amount of grain boundary (edge of grapheme sheet) in addition to intracrystalline defects, so determining which process is dominant in your setting becomes another problem to be addressed.

Response:
As we reported in a previous referee`s response, we attempted to avoid grain boundaries as much as possible, which was easily done due to the fact that the laser spot size (412 nm) was much smaller than the graphite grains in our samples (>10 microns, fig 4b). We acknowledge that some of the spectra may have been affected by an increase in the grain boundary density, however occasional measurements of this sort are unlikely to affect the average R2 per sample. Thus, we believe that the detected increase in D bands in our experimental data reflects disorder of the internal structure of graphite rather than grain size reduction. Nevertheless, a discussion addressing this topic is added in lines 225-233.

 I could faintly see very small platelets of graphite (< 1µm) in your photograph (Figure 4e), but damages during shin-section making also make this kind of roughness. I think it is better to provide high-resolution SEM images of the slip surface if you could not make good thin section.

Response:
 (1) We collected Raman spectra directly from the top of the sheared graphite gouges (referred to as shiny surfaces in the manuscript) to avoid damage induced during thin section making (the thin sections were made from other parts of the preserved experimental samples). High resolution SEM images of these surfaces (Fig. 4a and b) were also collected directly from the tops of the layers (labelled as XY sections in the manuscript).
(2) We imaged the zone underlying the shiny surfaces by cutting thin sections perpendicular to them. Thus, the small platelets of graphite (< 1µm) in Figure 4e, that you refer to, might have been affected by sample preparation. However, Raman spectra were not obtained from these samples, and thus the reported D bands do not reflect any damage induced during thin section making.

On the other hand, I think you should mention that the friction still remains low and stable if you applied shear strain >40. This feature may suggest the graphite on the slip surface still maintain its crystal perfection. In that sense, increments of R2 value of sheared graphite attributable mainly to grain size reduction but not amorphization.

Response:
Friction coefficients remain low and stable throughout all experiments. This is reflected in Fig. 1 and in lines 185-186. Our microstructural data clearly indicate partial structural disorder of the graphite structure, so we don't think that the reduction in the Raman spectra can be reasonably attributed just to grain size reduction. Also, please note there is no evidence of complete amorphization – graphite in all experiments remains crystalline even though significant defects in the graphite structure were introduced.

**Specific comments:**

Reviewer`s comment:
Line 46; I would happy if you add Oohashi et al. (2013, JGR) in the reference.
Response: The citation has been added into the reference list (in lines 378-380). Thank you for reminding us to properly acknowledge this excellent previous study.

Reviewer`s comment:
Line 60; I think initial thickness of gouge layer varies depends on applied normal stress if you put same weight of graphite powder for each experiments (becomes thinner under high normal stress). Did you change amount of graphite for normal stress of 5 MPa and 25 MPa to ensure to form exactly the same 3-mm thickness? This question is arises from why large shear strain is calculated from the experiments under 25 MPa normal stress (off course, I understand your explanation about compaction). Additionally, I would suspect thinning due to leakage of the gouge took place under 25 MPa experiments, especially for slip rate of 100 µm/s (Exp.10) because the gouge thickness seems to became less than half of the initial thickness.

Response: The mass of graphite was not changed throughout the various experiments and hence we acknowledge the effect of normal stresses on the attained shear strain in lines 190-192 (as you have also noted in your comment). Your concern about potential leakage of gouge material during Exp. 10 is based on the dramatic layer thinning recorded at the end of this experiment. However, figure 1b clearly shows a linear trend of increase in shear strain (based on layer thinning) with increase of sliding velocities in the experiments under normal stresses of 25 MPa (ranging from 21,45 through 31.86 to 46,77 in Exp. 8, 9 and 10 respectively). Therefore, we believe that leakage of gouge material is unlikely to have affected the measured layer thickness.

Reviewer`s comment:
Line 109-110; The authors documented µss does not depend on slip rates, and it remains constant for all experiments. However, I see clear relationship between µ at d=14-20 mm and slip rates; µ decreases with decreasing slip rates for σn=5 MPa, and µ decreases with increasing slip rates for σn=25 MPa.

Response: Dependence of µss on slip rates is not suggested by our data (Table 1). Instead we observe slight variations in 2 of the performed experiments (Exp. 4 and 10). This is reflected in the manuscript in lines 119-120.

Reviewer`s comment:
Line 141-143; The authors explain graphite crystalinity decreases with increasing slip rates for samples sheared under _n=25 MPa, and no slip rates dependence is found for samples sheared under _n=5 MPa. However, as you concluded, increase of R2 value can be attributed to applied shear strain but not to slip rates. I think you can not discuss direct relationship between R2 value and slip rates unless you conduct various slip-rates experiments at exactly the same shear strain.

Response: In lines 151-153, we simply compare the observed decrease in graphite crystallinity with the conditions of the experiments but do not imply direct relationship between R2 and slip rates. Then (lines 153-154), we conclude that 'graphite appears as most disordered in the experiments where the highest shear strain was achieved'.

Reviewer`s comment:
Line 181; I would suggest referring Di Toro et al. (2011, Nature) instead of Nakatani (2001).

Response: Thank you for the suggestion. The reference is now updated. (line 198)

Reviewer`s comment:
Table 2 and Figure 3 Please add errors and error bars for R2 value.

Response: Error estimates are now added in table 2 and line 432.

Hope this helps,
Kiyokazu Oohashi, Yamaguchi University, Japan

[revised manuscript text omitted]

---

## Referee Report (RR1)

Comments on se-2017-74-manuscript-version3
"Structural disorder of graphite and implications for graphite thermometry"

The revised version of the manuscript corrects some of the issues, which had been raised by previous reviews by R. Kilian and Oohashi Kiyokazu. However, the main points that had been criticised (interpretation of R2, sample deformation, R2 relation with deformation) and which are essential for the key message of the contribution remain not satisfactorily addressed at all. The authors add additional data or data treatment which, as outlined below does not necessarily help the authors to strengthen their argument against the criticism raised by the reviewers.

In short, the main issues were and still remain to be: 1) increase in R2 needs not to be equivalent to a decreased graphite structural order. 2) The "correction" for shear strain is actually not a correction and the relation of bulk sample strain to deformation which is actually resolved in the slip zones (see below) remains unclear. There are minor issues related to (a) the speculation on deformation mechanisms , (b) the way strain is treated and (c) how the total frictional work was calculated.

The message of the study seems to be that the graphite Raman thermometer is not applicable as-is in deformed rocks. However, the additional interpretations that go into the manuscript (e.g. effect of pressure and sliding velocity, deformation mechanism of graphite in these experiments) are not very well backed by the data and only deflect from the key message.

I recommend that the main issues need to be addressed to make this manuscript a robust publication. The experimental data (at best the raw data, displacement shear-stress, and including sample thickness data) and the Raman data are valuable to provide an insight into the great weaknesses of this Raman thermometry method and to start to understand the influence of deformation on the R2 ratio. However, the partly insufficient data treatment and presentation (e.g. experimental data, bulk strain estimates and sample thinning vs deformation in slip zones, or for example claims on grain sizes and microstructures in the text which are not in accordance with what the figures show) or the speculative interpretation (claims on deformation mechanism and R2 origin) is not what this kind of data has deserved. I am sure the authors could do better, also in cutting short in the speculations and strengthening the analytical parts by more coherent interpretations and a better description of methods for example.

1) Origin of reduced R2: The authors claim in numerous passages of the manuscript to report a reduced crystallinity (and it seems they define crystallinity by the presence of intragranular defects, or crystallographic perfection) of graphite. However, what they actually show is a ratio of peaks obtained from Raman spectroscopy (R2). The increase of R2 may be associated with increased intragranular defects or a reduced grain size (as already mentioned in the two previous reviews). It is further claimed that grain size reduction cannot account for the observed change in R2 - a claim the authors do not provide any evidence for. An increase of intergranular lattice defects is pure speculation and actually, newly provided data in the form of TEM images shows the opposite and are not in accordance to what the authors write in the manuscript. While stating that Raman measurements are obtained from within grains (which are said to be >10 μm - where does this number come from, Figure 4b? I'd strongly disagree that any grain size is evident from this Figure and if so, that 10 μm may be the lower limit), TEM images of the slip surface from where the Raman spectra were obtained show nanometer scale grains. By providing the TEM data, the authors actually contradict what they write in the text, as the data nicely shows that grain boundaries are produced and grain size is largely reduced with grains at the scale of several 10s of nm.

It is nicely demonstrated is that within slip surfaces, R2 is increased and the grain size is very small. What is not shown anywhere is that the increased R2 is related to an increase of intragranular defects - but exactly this is stated throughout the manuscript.

To overcome these ambiguities, providing a definition of crystallinity might be helpful. A crystal may have a low crystallinity because of a high density of (intragranular) lattice defects, however alternatively, an aggregate of "perfect" but extremely small grains may have a low crystallinity since there is a considerable volume which may be influenced by the disorder induced by grain boundaries. The latter definition would go very well with the presented data. Still, with the available data, nothing can be said about the intragranular defect density.

2) To correct the graphite thermometer for strain or not:
The authors present a fitted function (with surprisingly many digits) for R2 as a function of bulk shear strain or total frictional work. While they correctly note that any sort of strain-related correction will most likely never be feasible in nature (since it is most likely impossible to determine the amount of deformation within a specific graphite aggregate) I do not understand how this fitted function will be a "correction" for the thermometer. Shouldn't a correction provide the "true" R2 after removing the effect of deformation? That's not what the presented functionality provides.

3) The authors report that most of the strain is accommodated in thin slip surfaces. The variation of apparent bulk shear strain is due to sample thinning (unclear whether extrusion or compaction) so I would not expect this to principally affect the shear surface thickness since they seem to be consist of already highly compacted material. Hence it is pretty surprising to see a relation with the apparent bulk shear strain and R2. Using the mode of R2 instead of the mean, this relation actually becomes not so clear anymore. What is actually the reason to use the mean R2 and not the mode (e.g. by assuming that the most likely result will be analysed) of R2? Even when one would want to assume a relation between bulk strain and strain in slip zones, the chosen measure for strain is at most one of apparent shear strain, and it is not a 1:1 relation with the deformation within the sample. providing shear stress - displacement and displacement - thickness curves may already help with the interpretation of the data.

4) Calculation of total frictional work:
The authors included as an alternative measure to strain the total frictional work, given by the shear stress integrated over the displacement. I cannot reproduce the results presented in Figure 3 or Table 2 using the data of the authors shown in Figure 1a. Also, just by visual inspection, I do not see where the large differences between e.g. experiment8, 9 and 10 may arise from.
Unfortunately for experiments 2, 3, 6,7 no displacement - friction coefficient data was provided. I recommend to double check the calculation procedure for the frictional work  and to supply for all experiments displacemnt-shear stress curves. Additionally, it would be beneficial to provide also to provide the displacement-thickness relationships, such that the data could be adequately interpreted.

Following some additional comments with reference to the text:

L57: maximum grain size of 160μm : the maximum is in general a very unfortunate measure to say something about a material.
L74/75: ".. and summing." summing over?

I'd also still argue that the measure calculated by the authors is not a shear strain by it's definition in the sense that no strain ellipsoid could be derived from it and it is at best an apparent shear strain. The larger the thinning of the sample, the less related this number is to the deformation of the material. To describe the amount bulk deformation within the material e.g. the aspect ratio of the strain ellipsoid would be more meaningful. However, also for any of these consideration, it needs to be understood whether the sample is loosing volume (is compacting) or extruded somewhere.

Same for Section 3.1.2: While it is stated that the shear strain increase towards higher displacement velocities is basically an artefact of sample thinning (unclear whether extrusion or compaction), this fact is largely neglected in the rest of the ms. However, it not clearly noted that the choosen measure of deformation is with increasing sample thinning increasingly unrelated to the amount of deformation within the sample.

L84: Note that the laser spot size is not 1:1 equal with the Raman spot.

L141/142: "The degree of crystallinity in each sample..." Statement only makes sense when crystallinity would be defined as per aggregate - adding the option that grain boundary density increase decreases the overall crystallinity.

L147: "..crystallinity varies within each sample see above and speculation, first of all it is R2 which is variable.

L151/52: "Furthermore,...." a) table shows R2 not crystallinity as apparently defined by the authors. b) R2 actually increases (!) with increasing sliding velocities.

L156: work not "force" - also please check calculation procedure.

L166/167resp. Fig. 4b: grains of 10-50 μm size: unclear if grains or aggregates? Even if these are grains, it does not look like 10 μm is the lower limit. Also compare with the TEM images or Fig. 4d,e
Overall, entire section 3.3.1 does not convincingly demonstrate the large, 10 μm grains within the slip zones.

Section 3.3.2: Figure 5.a if grains are just a few 50-150 nm thick, how would measurements not be covering also grain boundaries. Kinking in graphite (if it is not twinning, but most angles seem larger than the typical twinning relation), Fig. 5b,c requires interlayer slip - and in case this is not crystallographically controlled will result in (001) parallel boundaries, and (001) perpendicular boundaries at kink boundaries, so this is another nice evidence for an increased boundary density beyond the Raman measurement scale. Fig. 5d: I'm not sure what I am seeing but are the authors sure this isn't already beam damaged material?

L200ff: "...shear strain variation systematically related to the condition of the experiments...shear strain is directly dependent on the applied normal stress". again, the value the authors calculate as shear strain is not a 1:1 measure for the amount of deformation within the material and an artefact of sample thinning.

Section 4.2 Structural disorder of graphite
largely this section should talk about R2 and what was really seen,. e.g. in the TEM images. it is NOT demonstrated that highly crystalline graphite is transformed into disordered graphite with strain!

What can actually be said is that large annealed grains show a low R2 and the deformed material has a (nano)scale grain size, increase of boundaries (grain boundaries, tilt/kink boundaries, ...) and a high R2. It's not possible to say something about crystallinity at the grain scale in the sense of intracrystalline defects.

L223: "...the results overall validate that structural disorder of graphite can result from shear deformation..." It is close to extremely enigmatic to me how this could be more than a speculation and how the obvious grain boundary area increase is totally ignored.

L229:230: How is it possible to say that no grain boundaries were measured if a) it is not clear where exactly the measurements were undertaken (see question and request from my first review) and b) TEM images of the slip surface show grain boundary spacing at the nm scale and c) the 10 µm grain size remains a speculation (it's totally unclear to me where this number comes from)? Is that some measurement or eyeballing?

L233:30: "...to disorder of the internal structure of graphite rather than grain size reduction." Please see above. This is not consistent with the images you provide!

L241: "...proven..." a) not a proof , b) not shear strain

L249: "We demonstrate that during shearing higher normal stress results in increased shear strain" No. And if layers thin just by compaction (volume reduction) I'd call the latter the reason, not a higher normal stress.

L257:"...effects of shear strain and pressure..." a) if anything at all, the only measure investigated was a bulk apparent shear strain and not pressure but normal stress. Depending on confinement, the pressures my vary of course, but I don't see how to derive/separate a pressure effect from that. Especially since the deformation of graphite seems to happen in very thin slip zones.

L269ff: "...fractured grains...", "brittle processes operated during shearing..resulted in structural disorder of graphite". While fracturing is for example and certainly intimately related to dislocations, it inevitably creates grain boundaries! A more thorough discussion on graphite deformation mechanism might also be more helpful.

L273: "...would not induce temperatures high enough fro crystal plastic processes" What are those for graphite? And processes such as?

L276: The authors probably mean crystal plastic mechanisms.

L277: Plastic deformation? It should be ductile deformation.

L281: "The crystallographic structure measured by Raman..." No, D1,D2 peaks are measured which could be interpreted in certain ways, e.g. related to structural state of a crystal lattice or e.g. grain boundary density, density of impurities... .

L285: "...mechanical modification of the graphite structure, which this study has identified..." No, the authors have identified an effect on R2, not directly on the graphite structure.

L286: " in deformed rocks" misleading, no rocks here beyond a pure graphite gouge.

L299: "...we propose a appropriate adjustment based on our dataset" I don't find - beyond my and the authors doubts on a useful applicability of such an adjustment - any suggestion how this adjustment should look like.

L307: "Furthermore, it can be challenging to estimate shear strain in nature ..." Yes it can be challenging and it will be even more challenging to translate bulk rock strain to a deformation seen by a particular grain of graphite within a deformed rock.
It should actually also be noted that in the experiments it does seem challenging to estimate the true shear strain/deformation in the bulk: and even more challenging to estimate the deformation in the actually deforming layer of graphite and this is what would be required to start any correction at all.

L310: "... graphite crystallinity.." use R2 instead of crystallinity unless properly defined

L312: ""...graphite structural order" see above, use R2

L313: "Microstructural data reveal that this is a result of brittle processes." This needs to be clearly laid out in the results and in the discussion.

L314: " trend of increasing shear strain as a function of normal stress and sliding velocity..." this is an effect of sample thinning. And not that this shear strain is does not 1:1 relate to deformation seen by the bulk sample.

L318: "...simple shear strain calibration.." a) there is no such thing as "simple shear strain", this is nonsense b) there seems to be substantial thinning of the samples, and while incorrectly treating it as simple shear to calculate an apparent shear strain, the data does not relate to simple shear flow.

Rüdiger Kilian

---

## Referee Report (RR2)

Review of SE-2017-44 version 4

The manuscript has many improvements compared to the previous versions and is suitable for publication after minor revisions/corrections, mostly being of technical nature.

A few comments:
1) While the authors acknowledge now that the measured spectra do not necessarily display graphite crystallinity (in the sense of intragranular defects) but an aggregate crystallinity, the manuscript will further benefit from a very brief explanation or definition of crystallinity as now used by the authors. So it should be made clear what the authors mean since I think this is not clear for every reader. Also, I'd recommend to make sure that this definition is kept throughout the manuscript and consistently used.

2) The discussion could be benefit from a few sentences actually discussion the nature of D1 and D2 bands (intralayer and edge defects) and how they may relate to deformation and grain size decrease. The difference and nature of D1 and D2 bands could also follow on L91.

3) Please also provide the shear stress-displacement curve that appears in the authors response. This is the measured (raw) data and should be provided.
Also, please make sure that the curves are readable, no double colors, maybe work with dashed lines or similar tricks so it's easier to find the curves for each experiment.

L74: apparent shear strain
L95: Are you sure this is the Y-Z sections? when you look onto the sliding surface I'd say it's X-Y.
L109: what was observed are Raman spectra, not structural modifications, unless this refers to structure as in microstructure.
L164; oriented nearly paralle lto the shear directions What should that mean? You probably mean 001 of graphite grains are parallel to the shear plane?
L167:"...aligned with 001 parallel to the slip direction" - I guess this should be plane not direction
L178: "...with random orientations" How is that meant? Grain shapes, grain (crystal) orientations?
 L207: Tunistra and Koening, 19780 missing in references
L227:  replace "shear strain" with deformation or similar
L230: "structural order" should be aggregate crystallinity or the term that the authors need to define to prevent these speculations
L244: insert "and"
L272: "structural disorder" should be aggregate crystallinity or the term that the authors need to define to prevent these speculations
L272ff: odd sentence, maybe make two out of it.

Figure captions to
Figure 1 and 3: use apparent shear strain
Figure 4: which one is BSE/SE
L367: Fig 4b "well compacted layer of aligned grains" - I cannot see this in the image
L370: ...oriented sub-perpendicular to the shear direction" What should that mean?
L389: "error estimate of..." Where does that appear?

Figure 5 has no figure captions with line numbers: "kink band stacking" What should that be?

Best regards,
Rüdiger Kilian

---

## Editor Decision (ED1)

Final technical corrections for se-2017-74

1. Please provide definitions or explanations for the following terms and concepts.
Just introduce a sentence or two when you first talk about them:
- 'graphite crystallinity' and 'aggregate crystallinity'
- the nature of and the difference between D1 and D2 bands

2. Tunistra and Koening, 1978, is missing in the references

3. Please correct in text:

line 178: '...with random orientations' specify if grain shapes, grain (crystal)
orientations.
line 227: replace 'shear strain' with, e.g., 'deformation'
line 230: replace 'structural order' by 'aggregate crystallinity'
line 244: insert 'and'
line 272:replace 'structural disorder' with 'aggregate crystallinity'
line 272ff: please rephrase, make two sentences

4. Please correct in figure captions:

Figure 1 and 3: replace 'shear strain' by 'apparent shear strain' (fix reference on line
74)
Figure 2: where is the description of the 'error estimate'
Figure 4: indicate if BSE or SE contrast
Figure 4b: should be 'X-Y' section, not Y-Z (fix reference in text line 95)
Figure 4e: why 'sub-perpendicular to the shear direction'? shouldn't it be the shear
plane? and why sub-perpendicular. Do you mean 001 of graphite grains are parallel
to the shear plane?
Please clear up caption and corresponding text (line 164 to 169).
Figure 5 clarify what you mean by 'kink band stacking'

---

## Author Response (AR2)

Dr. Renée Heilbronner
Handling Topical Editor, Solid Earth

Dear Renée Heilbronner,

Please accept our revised version of the manuscript entitled 'Structural Disorder of Graphite and Implications for Graphite Thermometry' for consideration for publication in Solid Earth. We agree with you that the main message of the manuscript is that the calibrated graphite thermometer is unreliable in deformed rocks, and we hope we have outline this aspect well enough throughout the text. However, we believe it is important to understand why we observe changes in the Raman spectra of graphite after shear deformation, and thus we discuss and interpret the estimated R2 ratios with respect to the parameters of the experiments and our microstructural data. We hope you will agree to retain this discussion.

Nevertheless, we have carefully modified the manuscript in accordance to reviewer`s recommendations and we have implemented the requested changes. We have re-written most of the discussion for better clarity, in particular the parts that interpret (1) shear strain estimates (2) changes in the R2 values and (3) correlation between shear strain and average R2. Also, we agree that the previously suggested shear strain calibration of the graphite thermometer may be inapplicable in natural rocks, and thus we have removed this part from the manuscript. Furthermore, we have added some additional plots showing our mechanical data (included in figure 1), and we also re-calculated the total frictional work per each experiment. Apart from this, we were willing to include the raw mechanical data as well, however the dataset for a single experiment is over a thousand pages, and thus we find it unessential to include. However, we are happy to provide a link to this data to a reader interested in more details.

Below we included the reviewer`s comments and then provided detailed responses to all of them (in red colour). We refer to the revisions of the manuscript with line numbers that are correct with respect to this version of the manuscript with changes accepted.

We hope that the revised manuscript meets the requirements of the high-quality journal Solid Earth, and it will be considered for publication.

Regards,
Martina Kirilova
Corresponding author
Martina.a.kirilova@gmail.com

On behalf of the authors: Virginia Toy, Jeremey S. Rooney, Carolina Giorgetti, Keith C. Gordon, Cristiano Collettini and Toru Takeshita

**Comments on se-2017-74-manuscript-version3**
**"Structural disorder of graphite and implications for graphite thermometry"**

The revised version of the manuscript corrects some of the issues, which had been raised by previous reviews by R. Kilian and Oohashi Kiyokazu. However, the main points that had been criticised (interpretation of R2, sample deformation, R2 relation with deformation) and which are essential for the key message of the contribution remain not satisfactorily addressed at all. The authors add additional data or data treatment which, as outlined below does not necessarily help the authors to strengthen their argument against the criticism raised by the reviewers.

In short, the main issues were and still remain to be: 1) increase in R2 needs not to be equivalent to a decreased graphite structural order. 2) The "correction" for shear strain is actually not a correction and the relation of bulk sample strain to deformation which is actually resolved in the slip zones (see below) remains unclear. There are minor issues related to (a) the speculation on deformation mechanisms, (b) the way strain is treated and (c) how the total frictional work was calculated.

The message of the study seems to be that the graphite Raman thermometer is not applicable as-is in deformed rocks. However, the additional interpretations that go into the manuscript (e.g. effect of pressure and sliding velocity, deformation mechanism of graphite in these experiments) are not very well backed by the data and only deflect from the key message.

I recommend that the main issues need to be addressed to make this manuscript a robust publication. The experimental data (at best the raw data, displacement shear-stress, and including sample thickness data) and the Raman data are valuable to provide an insight into the great weaknesses of this Raman thermometry method and to start to understand the influence of deformation on the R2 ratio. However, the partly insufficient data treatment and presentation (e.g. experimental data, bulk strain estimates and sample thinning vs deformation in slip zones, or for example claims on grain sizes and microstructures in the text which are not in accordance with what the figures show) or the speculative interpretation (claims on deformation mechanism and R2 origin) is not what this kind of data has deserved. I am sure the authors could do better, also in cutting short in the speculations and strengthening the analytical parts by more coherent interpretations and a better description of methods for example.

Response: Yes, the key message is that the Raman thermometry is not applicable in deformed rocks, but we think it is very important to understand why and thus we provide additional data. We have clarified our interpretation in accordance with reviewer's recommendation and we hope you will agree to retain this discussion.

1) Origin of reduced R2: The authors claim in numerous passages of the manuscript to report a reduced crystallinity (and it seems they define crystallinity by the presence of intragranular defects, or crystallographic perfection) of graphite. However, what they actually show is a ratio of peaks obtained from Raman spectroscopy (R2). The increase of R2 may be associated with increased intragranular defects or a reduced grain size (as already mentioned in the two previous reviews). It is further claimed that grain size reduction cannot account for the observed change in R2 - a claim the authors do not provide any evidence for. An increase of intergranular lattice defects is pure speculation and actually, newly provided data in the form of TEM images shows the opposite and are not in accordance to what the authors write in the manuscript. While stating that Raman measurements are obtained from within grains (which are said to be >10 μm - where does this number come from, Figure 4b? I'd strongly disagree that any grain size is evident from this Figure and if so, that 10 μm may be the lower limit), TEM images of the slip surface from where the Raman spectra were obtained show nanometer scale grains. By providing the TEM data, the authors actually contradict what they write in the text, as the data nicely shows that grain boundaries are produced and grain size is largely reduced with grains at the scale of several 10s of nm.

It is nicely demonstrated is that within slip surfaces, R2 is increased and the grain size is very small. What is not shown anywhere is that the increased R2 is related to an increase of intragranular defects - but exactly this is stated throughout the manuscript.

To overcome these ambiguities, providing a definition of crystallinity might be helpful. A crystal may have a low crystallinity because of a high density of (intragranular) lattice defects, however alternatively, an aggregate of "perfect" but extremely small grains may have a low crystallinity since there is a considerable volume which may be influenced by the disorder induced by grain boundaries. The latter definition would go very well with the presented data. Still, with the available data, nothing can be said about the intragranular defect density.

Response: Yes, we agree with this comment and we admit that our discussion needed clarification. Thus, we modified section 4.2 accordingly and we clearly acknowledged the effect of increase in grain boundary density on Raman spectra of graphite. Now, we also use the term 'aggregate crystallinity' to avoid any confusion with intragranular crystallinity. (lines 202-212)

2) To correct the graphite thermometer for strain or not: The authors present a fitted function (with surprisingly many digits) for R2 as a function of bulk shear strain or total frictional work. While they correctly note that any sort of strain-related correction will most likely never be feasible in nature (since it is most likely impossible to determine the amount of deformation within a specific graphite aggregate) I do not understand how this fitted function will be a "correction" for the thermometer. Shouldn't a correction provide the "true" R2 after removing the effect of deformation? That's not what the presented functionality provides.

Response: We previously acknowledged that the suggested calibration may be impossible to use in natural rocks. We agree that suggesting a calibration that is perhaps not applicable is pointless, thus we have removed this part from the text.

3) The authors report that most of the strain is accommodated in thin slip surfaces. The variation of apparent bulk shear strain is due to sample thinning (unclear whether extrusion or compaction) so I would not expect this to principally affect the shear surface thickness since they seem to be consist of already highly compacted material. Hence it is pretty surprising to see a relation with the apparent bulk shear strain and R2. Using the mode of R2 instead of the mean, this relation actually becomes not so clear anymore. What is actually the reason to use the mean R2 and not the mode (e.g. by assuming that the most likely result will be analysed) of R2? Even when one would want to assume a relation between bulk strain and strain in slip zones, the chosen measure for strain is at most one of apparent shear strain, and it is not a 1:1 relation with the deformation within the sample. providing shear stress - displacement and displacement - thickness curves may already help with the interpretation of the data.

Response: We understand your concerns about the way we estimated shear strain and we agree that the presented values in fact represent apparent shear strain. Thus, we have included a clarification of this issue in lines 191-193.

We have used average R2 per sample in order to investigate any potential correlation with the overall structural disorder per sample with the conditions of the corresponding experiment. We clearly indicate in the discussion that the correlation between R2 and the estimated shear strain values is a rough approximation (lines 223-225).

4) Calculation of total frictional work: The authors included as an alternative measure to strain the total frictional work, given by the shear stress integrated over the displacement. I cannot reproduce the results presented in Figure 3 or Table 2 using the data of the authors shown in Figure 1a. Also, just by visual inspection, I do not see where the large differences between e.g. experiment 8, 9 and 10 may arise from. Unfortunately for experiments 2, 3, 6,7 no displacement - friction coefficient data was provided. I recommend to double check the calculation procedure for the frictional work and to supply for all experiments displacemnt-shear stress curves. Additionally, it would be beneficial to provide also to provide the displacement-thickness relationships, such that the data could be adequately interpreted.

Response: Thank you for your comment. We checked our calculations of total frictional work and discovered our mistake. The new values are now shown in Table 2. We re-calculated frictional work by using trapezoidal estimation on Matlab, then plotting the resulting data and fitting a power2 function to that.

Below is a snippet from the matlab script that calculates the frictional work:
```
    eval([filename '.FricWorkTrap = trapz(' filename '.ec_disp,' filename '.Tau);'])
```

Below is a snippet from the plotting of the total frictional work, X and Y are set with the values calculated above:
```
f = fittype('power2');
F = fit(X,Y,f)
txt1 = ['F(x) = ' num2str(F.a) 'x^{' num2str(F.b) '}+' num2str(F.c)];
plot(F,X,Y,'o')
```

We also added the displacement-friction coefficient plots for experiments 2,3,6 and 7 to figure 1a, as well as displacement- thickness plots for all experiments in figure 1c. Plots of displacement vs shear stress do not significantly contribute to the main message, and thus we chose not to include them in the manuscript. But because the reviewer asked for these plots, we provide them in the figure below:

[Figure]

Following some additional comments with reference to the text:

L57: maximum grain size of 160μm : the maximum is in general a very unfortunate measure to say something about a material.

We now present average grain size values (line 57).

L74/75: .".. and summing." summing over?

Replaced with 'summing up' (line 75)

I'd also still argue that the measure calculated by the authors is not a shear strain by it's definition in the sense that no strain ellipsoid could be derived from it and it is at best an apparent shear strain. The larger the thinning of the sample, the less related this number is to the deformation of the material. To describe the amount bulk deformation within the material e.g. the aspect ratio of the strain ellipsoid would be more meaningful. However, also for any of these consideration, it needs to be understood whether the sample is loosing volume (is compacting) or extruded somewhere.

Above we agreed that it is apparent shear strain and we have called it that in the text.

Same for Section 3.1.2: While it is stated that the shear strain increase towards higher displacement velocities is basically an artefact of sample thinning (unclear whether extrusion or compaction), this fact is largely neglected in the rest of the ms. However, it not clearly noted that the choosen measure of deformation is with increasing sample thinning increasingly unrelated to the amount of deformation within the sample.

We acknowledged this in the discussion. (lines 195 -198).

L84: Note that the laser spot size is not 1:1 equal with the Raman spot.

Thanks for the comment.

L141/142: "The degree of crystallinity in each sample..." Statement only makes sense when crystallinity would be defined as per aggregate - adding the option that grain boundary density increase decreases the overall crystallinity.

We now talk about R2 instead (line 141).

L147: "..crystallinity varies within each sample see above and speculation, first of all it is R2 which is variable.

We now say '… R2 values vary within a sample…' (line 145)

L151/52: "Furthermore,...." a) table shows R2 not crystallinity as apparently defined by the authors.

Okay, we discuss R2 now (line 149)

b) R2 actually increases (!) with increasing sliding velocities.

Yes, correct. And when R2 increases the crystallinity decreases, which is what we had in the text. The text is now modified, and we talk about increase in R2 instead. (line 149)

L156: work not "force" - also please check calculation procedure.

Thanks for noticing this mistake. We have recalculated frictional work (see comment 4).

L166/167resp. Fig. 4b: grains of 10-50 μm size: unclear if grains or aggregates? Even if these are grains, it does not look like 10 μm is the lower limit. Also compare with the TEM images or Fig. 4d,e. Overall, entire section 3.3.1 does not convincingly demonstrate the large, 10 μm grains within the slip zones.

We now say '…from < 5 to 10 micrometers…' (line 160). TEM data is discussed in the following section 3.3.2.

Section 3.3.2: Figure 5.a if grains are just a few 50-150 nm thick, how would measurements not be covering also grain boundaries. Kinking in graphite (if it is not twinning, but most angles seem larger than the typical twinning relation), Fig. 5b,c requires interlayer slip - and in case this is not crystallographically controlled will result in (001) parallel boundaries, and (001) perpendicular boundaries at kink boundaries, so this is another nice evidence for an increased boundary density beyond the Raman measurement scale. Fig. 5d: I'm not sure what I am seeing but are the authors sure this isn't already beam damaged material?

As we responded to a previous comment, we now acknowledge in the text the effects of increase in grain boundary density on Raman spectra.

L200ff: "...shear strain variation systematically related to the condition of the experiments...shear strain is directly dependent on the applied normal stress". again, the value the authors calculate as shear strain is not a 1:1 measure for the amount of deformation within the material and an artefact of sample thinning.

We removed this part of the discussion.

Section 4.2 Structural disorder of graphite. largely this section should talk about R2 and what was really seen,. e.g. in the TEM images. it is NOT demonstrated that highly crystalline graphite is transformed into disordered graphite with strain!

What can actually be said is that large annealed grains show a low R2 and the deformed material has a (nano)scale grain size, increase of boundaries (grain boundaries, tilt/kink boundaries, ...) and a high R2. It's not possible to say something about crystallinity at the grain scale in the sense of intracrystalline defects.

As we responded to a previous comment, we agree and we have modified section 4.2 accordingly.

L223: "...the results overall validate that structural disorder of graphite can result from shear deformation..." It is close to extremely enigmatic to me how this could be more than a speculation and how the obvious grain boundary area increase is totally ignored.

We now refer to 'graphite aggregates' instead. (line 218).

L229:230: How is it possible to say that no grain boundaries were measured if a) it is not clear where exactly the measurements were undertaken (see question and request from my first review) and b) TEM images of the slip surface show grain boundary spacing at the nm scale and c) the 10 μm grain size remains a speculation (it's totally unclear to me where this number comes from)? Is that some measurement or eyeballing?

We modified this part of the discussion.

L233:30: "...to disorder of the internal structure of graphite rather than grain size reduction." Please see above. This is not consistent with the images you provide!

We modified this part of the discussion.

L241: "...proven..." a) not a proof , b) not shear strain

L249: "We demonstrate that during shearing higher normal stress results in increased shear strain" No. And if layers thin just by compaction (volume reduction) I'd call the latter the reason, not a higher normal stress.

We removed this part from the discussion.

L257:"...effects of shear strain and pressure..." a) if anything at all, the only measure investigated was a bulk apparent shear strain and not pressure but normal stress. Depending on confinement, the pressures my vary of course, but I don't see how to derive/separate a pressure effect from that. Especially since the deformation of graphite seems to happen in very thin slip zones.

We removed this part from the discussion.

L269ff: "...fractured grains...", "brittle processes operated during shearing..resulted in structural disorder of graphite". While fracturing is for example and certainly intimately related to dislocations, it inevitably creates grain boundaries! A more thorough discussion on graphite deformation mechanism might also be more helpful.

Thank you for the comment.

L273: "...would not induce temperatures high enough fro crystal plastic processes" What are those for graphite? And processes such as?

We meant ductile processes. The text is now modified (line 239).

L276: The authors probably mean crystal plastic mechanisms.

Yes, thank you for the comment (line 242)

L277: Plastic deformation? It should be ductile deformation.

Replaced (line 243).

L281: "The crystallographic structure measured by Raman..." No, D1,D2 peaks are measured which could be interpreted in certain ways, e.g. related to structural state of a crystal lattice or e.g. grain boundary density, density of impurities... .

We now say 'The structural order of graphite measured by Raman spectroscopy' (line 251) (Beyssac et al, 2002, 2003).

L285: "...mechanical modification of the graphite structure, which this study has identified..." No, the authors have identified an effect on R2, not directly on the graphite structure.

We rephrased this to '....this thermometer disregards the effects of mechanical modifications on the structure of graphite aggregates, which this study has identified as having a substantial influence on the R2 ratios in deformed graphite gouges at sub-seismic velocities.' (lines 255-257)

L286: " in deformed rocks" misleading, no rocks here beyond a pure graphite gouge.

Replaced with 'deformed graphite gouges' (lines 255-257)

L299: "...we propose a appropriate adjustment based on our dataset" I don't find - beyond my andthe authors doubts on a useful applicability of such an adjustment - any suggestion how this adjustment should look like.

Removed from the text.

L307: "Furthermore, it can be challenging to estimate shear strain in nature ..." Yes it can be challenging and it will be even more challenging to translate bulk rock strain to a deformation seen by a particular grain of graphite within a deformed rock. It should actually also be noted that in the experiments it does seem challenging to estimate the true shear strain/deformation in the bulk: and even more challenging to estimate the deformation in the actually deforming layer of graphite and this is what would be required to start any correction at all.

Removed from the text.

L310: "... graphite crystallinity.." use R2 instead of crystallinity unless properly defined

We modified our conclusions according to the overall revision (lines 272-276)

L312: ""...graphite structural order" see above, use R2

We modified our conclusions according to the overall revision (lines 272-276)

L313: "Microstructural data reveal that this is a result of brittle processes." This needs to be clearly laid out in the results and in the discussion.

It is in lines 235-237.

L314: " trend of increasing shear strain as a function of normal stress and sliding velocity..." this is an effect of sample thinning. And not that this shear strain is does not 1:1 relate to deformation seen by the bulk sample.

Removed from the conclusions.

L318: "...simple shear strain calibration.." a) there is no such thing as "simple shear strain", this is nonsense b) there seems to be substantial thinning of the samples, and while incorrectly treating it as simple shear to calculate an apparent shear strain, the data does not relate to simple shear flow.

Removed from the conclusions.

Rüdiger Kilian

[revised manuscript text omitted]